# Midbrain dopaminergic inputs gate amygdala intercalated cell clusters by distinct and cooperative mechanisms in male mice

Ayla Aksoy-Aksel[1,2,3], Andrea Gall[1,2,3], Anna Seewald[4], Francesco Ferraguti[4], Ingrid Ehrlich[1,2,3]*

[1]Hertie Institute for Clinical Brain Research, Tübingen, Germany; [2]Centre for Integrative Neuroscience, Tübingen, Germany; [3]Department of Neurobiology, Institute of Biomaterials and Biomolecular Systems, University of Stuttgart, Stuttgart, Germany; [4]Department of Pharmacology, Medical University of Innsbruck, Innsbruck, Austria

**Abstract** Dopaminergic signaling plays an important role in associative learning, including fear and extinction learning. Dopaminergic midbrain neurons encode prediction error-like signals when threats differ from expectations. Within the amygdala, GABAergic intercalated cell (ITC) clusters receive one of the densest dopaminergic projections, but their physiological consequences are incompletely understood. ITCs are important for fear extinction, a function thought to be supported by activation of ventromedial ITCs that inhibit central amygdala fear output. In mice, we reveal two distinct novel mechanisms by which mesencephalic dopaminergic afferents control ITCs. Firstly, they co-release GABA to mediate rapid, direct inhibition. Secondly, dopamine suppresses inhibitory interactions between distinct ITC clusters via presynaptic D1 receptors. Early extinction training augments both GABA co-release onto dorsomedial ITCs and dopamine-mediated suppression of dorso- to ventromedial inhibition between ITC clusters. These findings provide novel insights into dopaminergic mechanisms shaping the activity balance between distinct ITC clusters that could support their opposing roles in fear behavior.

*For correspondence:
ingrid.ehrlich@bio.uni-stuttgart.de

**Competing interests:** The authors declare that no competing interests exist.

## Introduction

Mesencephalic dopaminergic neurons in the ventral tegmental area (VTA) and substantia nigra pars compacta (SNC) constitute a neuromodulatory system that has been linked to error prediction and salience coding (*Bromberg-Martin et al., 2010*; *Horvitz, 2000*; *Schultz et al., 1997*). Prediction error signals drive the formation and the updating of stimulus-outcome associations by detection of mismatches between actual and expected experiences (*Pearce and Hall, 1980*; *Rescorla and Wagner, 1972*), which also operate when conditioned stimulus (CS)-unconditioned stimulus (US) and CS-no US associations need to be formed during fear conditioning and extinction learning, respectively (*Bouton, 2004*; *McNally et al., 2011*; *Salinas-Hernández et al., 2018*). More specifically, omission of an expected aversive stimulus during the early sessions of extinction training activates a subset of VTA dopaminergic neurons (*Cai et al., 2020*; *Salinas-Hernández et al., 2018*). Consequently, optogenetic inhibition of the VTA dopaminergic neurons during extinction training impairs, and their excitation enhances the extinction learning (*Luo et al., 2018*; *Salinas-Hernández et al., 2018*).

The amygdala is a key structure that plays a critical role in mediating fear- and anxiety-related behaviors, and is a primary site for acquisition and storage of fear memory (*Davis, 2000*; *Duvarci and Pare, 2014*; *Luo et al., 2018*; *Salinas-Hernández et al., 2018*). Dopamine (DA) is

released in the amygdala during affective states, such as stress or fear (*Inglis and Moghaddam, 1999*; *Yokoyama et al., 2005*), and direct pharmacological intervention on dopaminergic receptors in the central amygdala (CeA) or basal amygdala (BA) affects the acquisition and storage of fear memories (*Guarraci et al., 1999*; *Lee et al., 2017*).

Interestingly, selective optogenetic stimulation of midbrain dopaminergic axons results in the release not only of the neuromodulator DA, but also of the fast-acting classical neurotransmitters GABA and/or glutamate in the dorsal striatum and nucleus accumbens (*Granger et al., 2017*; *Tritsch et al., 2016*). In the amygdala, glutamate was shown to be co-released together with DA in the CeA (*Groessl et al., 2018*; *Mingote et al., 2015*) and BA, in the latter, preferentially onto fast-spiking interneurons (*Lutas et al., 2019*). However, a co-release of GABA has not been reported so far.

Several lines of evidence suggest that within the amygdala, the intercalated cells (ITCs) could be a key target for dopaminergic regulation. ITCs constitute a specialized network of GABAergic cells and, in mice, are organized in several clusters around the basolateral complex of the amygdala (BLA) (*Busti et al., 2011*; *Geracitano et al., 2007*; *Marowsky et al., 2005*). Dopaminergic axons densely innervate the somata and dendrites of ITCs (*Asan, 1997*; *Fuxe et al., 2003*); however, the functional consequence of this innervation remained largely unexplored. Furthermore, ITCs show a high expression level of pre- and postsynaptic DA receptors (*Fuxe et al., 2003*; *Pinto and Sesack, 2008*; *Wei et al., 2018*). DA application in brain slices directly hyperpolarizes neurons in the medial and lateral ITC clusters via D1 receptors (DRD1) and thereby suppresses their output to CeA and BLA, respectively (*Gregoriou et al., 2019*; *Mańko et al., 2011*; *Marowsky et al., 2005*).

In that respect, ITC clusters are ideally positioned to integrate dopaminergic signals with the sensory information that is either conveyed directly (*Asede et al., 2015*; *Barsy et al., 2020*; *Strobel et al., 2015*) or that has been preprocessed in the BLA (*Herry and Johansen, 2014*; *Kwon et al., 2015*; *Paré et al., 2004*). Indeed, ITCs and their plasticity have been shown to play a significant role in extinction (*Amano et al., 2010*; *Likhtik et al., 2008*) and, more recently, also in fear learning and memory (*Asede et al., 2015*; *Busti et al., 2011*; *Huang et al., 2014*; *Kwon et al., 2015*). The classical view on ITC function within amygdala circuits posits an inhibitory action onto neighboring CeA and BLA nuclei to gate information flow (*Ehrlich et al., 2009*; *Marowsky et al., 2005*; *Morozov et al., 2011*; *Paré et al., 2004*). However, there is emerging evidence pointing toward a more complex picture that involves temporal separation in the activity of the individual clusters, i.e., dorsomedial (dm)- and ventromedial (vm)-ITC clusters, in fear recall and extinction (*Busti et al., 2011*). Considering the connections between the different clusters (*Asede et al., 2015*; *Busti et al., 2011*; *Duvarci and Pare, 2014*; *Royer et al., 2000*), it is plausible that ITCs' inhibitory actions, not only onto CeA and BLA, but also onto other ITC clusters, could be modulated by DA.

Thus, while mounting evidence supports that midbrain dopaminergic inputs play important roles in controlling amygdala function in fear learning as well as extinction (*Abraham et al., 2014*; *Lee et al., 2017*), the cellular impact onto specific amygdala microcircuits is incompletely understood (*Grace et al., 2007*; *Lee et al., 2017*). To address this knowledge gap, we used specific optogenetic stimulation of dopaminergic axons from VTA/SNC to explore the functional impact onto ITCs. We observed co-release of GABA from dopaminergic axons onto ITCs mediating fast inhibition. Phasic stimulation of dopaminergic fibers suppressed spontaneous inhibitory inputs onto dm-ITCs in a DA receptor-dependent manner. Consequently, when interrogating transmission between ITC clusters, we observed a presynaptic depression mediated by DRD1. Extinction learning promoted the co-release of GABA onto dm-ITCs and the DA-induced suppression of inhibition between dm- and vm-ITC clusters. Our results demonstrate a dual mechanism of action of dopaminergic inputs, with cooperative fast ionotropic and slower metabotropic components, which jointly regulate the inhibitory network formed between ITCs. We suggest that this can tip the activity balance between individual clusters to support fear suppression.

## Results

### Dopaminergic fibers from VTA/SNC innervate ITC clusters in amygdala

We first examined the projections of midbrain dopaminergic inputs to amygdala ITC clusters and surrounding amygdala regions. To this end, we targeted dopaminergic neurons selectively using a

DAT-Cre mouse line and injected a Cre-dependent recombinant adenoassociated virus (AAV) encoding ChR2-YFP into the VTA/SNC (*Figure 1—figure supplement 1A*). Our injections resulted in expression of ChR2-YFP mostly in VTA and to a lesser extent in SNC neurons, the vast majority of which were immunopositive for tyrosine hydroxylase (TH) (*Figure 1—figure supplement 1B–C*). We next examined YFP+ fibers in the amygdala (*Figure 1—figure supplement 2A*). YFP+ fibers targeting FoxP2-positive neurons in the dm- and vm-ITC clusters were also immunoreactive for TH, indicating that they are dopaminergic in nature (*Figure 1A–B*). In line with previous reports (*Lutas et al., 2019*; *Mingote et al., 2015*), we also observed YFP+ fibers co-localized with TH-labeled axons in the CeA, and more sparsely in the BA (*Figure 1—figure supplement 2B–C*).

In depth analysis of VTA/SNC inputs onto dm- and vm-ITC clusters revealed that ChR-YFP+ afferents were more prominent in the dm- compared to the vm-ITC cluster (*Figure 1C–D*, *Figure 1—figure supplement 3*). Because these results could be confounded by size and localization of midbrain injection sites, we also compared the density of TH+ fibers in wild-type animals. In keeping, the density of TH+ fibers was also higher in the dm-ITC versus vm-ITC cluster at bregma levels comparable to those from 3D reconstructions, but not at the most caudal levels of the amygdala (*Figure 1—figure supplement 4A–C*). The differential innervation of the dm-ITC versus vm-ITC clusters could at least in part be due to distinct patterns of afferent projections from VTA and SNC. Dual color labeling with viral injections biased toward medial VTA and lateral VTA/SNC indicated that dm-ITCs receive afferents from both VTA and SNC, whereas vm-ITCs are mostly targeted by VTA (*Figure 1—figure supplement 5A–B*). Taken together, this suggests that dm- and vm-ITC clusters are differentially innervated by the dopaminergic midbrain, both in terms of density and in terms of source of their afferents.

We next asked if YFP+ midbrain afferents made synaptic contacts with neurons in the dm- and vm-ITC clusters. To this aim, we filled ITCs with biocytin, which allowed us to visualize their somato-dendritic domain. We then labeled putative presynaptic active zones by immunostaining for Bassoon (*Liu et al., 2018*). Co-localization of Bassoon and ChR2-YFP was observed in close proximity to dendrites and somata of ITCs in dm- and vm-ITC clusters (*Figure 1E–F*, white arrows), suggesting that DA midbrain inputs make functional synaptic contacts.

Midbrain dopaminergic neurons from VTA and SNC co-release GABA and/or glutamate in the dorsal striatum and nucleus accumbens (*Granger et al., 2017*; *Mingote et al., 2015*; *Stuber et al., 2010*; *Tecuapetla et al., 2010*; *Tritsch et al., 2012*; *Tritsch et al., 2016*). In the amygdala, functional co-release of glutamate from dopaminergic VTA afferents has been demonstrated in CeA (*Mingote et al., 2015*) and, to a lesser extent, in the BA (*Lutas et al., 2019*). To address whether this may also be the case for ITCs, we first checked for the presence of vesicular GABA or glutamate transporters in ChR2-YFP+ axons. We indeed observed immunoreactivity for the vesicular glutamate transporters vGluT1/2 and vesicular GABA transporter vGAT in a subset of ChR2-YFP+ presynaptic boutons within FoxP2-positive dm- and vm-ITC clusters (*Figure 1G–J*), corroborating the hypothesis of GABA and glutamate co-release from midbrain dopaminergic fibers. Because of incomplete penetration of the vGluT1/2 and vGAT antibodies into slices, a quantitative analysis of the co-localization with ChR2-YFP+ terminals was not carried out.

## Dopaminergic fibers co-release mainly GABA in the medial ITC clusters

In light of our anatomical observations of a possible co-release of GABA or glutamate, we explored whether optical stimulation of ChR2-YFP+ fibers from confirmed dopaminergic midbrain injections (*Figure 2—figure supplement 1A–C*) evoked fast postsynaptic currents (PSCs) in dm- and vm-ITCs. Indeed, we could detect PSCs in the majority of the neurons (62–84%) recorded in whole-cell patch-clamp mode in ITC clusters as well as in the CeA (*Figure 2A–B,D,G*), but not in principal neurons of the BA (10 neurons from five animals showing co-release in either CeA or ITCs, *Figure 2—figure supplement 2D*).

To classify fast PSCs according to neurotransmitter type, we first determined their reversal potential (*Figure 2E and H*). In a large fraction of neurons within dm- and vm-ITC clusters, we found PSCs that reversed close to the calculated reversal potential for chloride (−47.3 mV), suggesting that they were GABAergic (*Figure 2C,E and H*). Another fraction of neurons displayed PSCs with more depolarized reversal potentials (>-40 and <-15 mV), suggesting that they were mixed PSCs, and finally, a small fraction of dm-ITCs displayed PSCs that reversed close to 0 mV, suggesting that they were glutamatergic (*Figure 2C,E and H*). To ascertain our classification, we identified the neurotransmitter

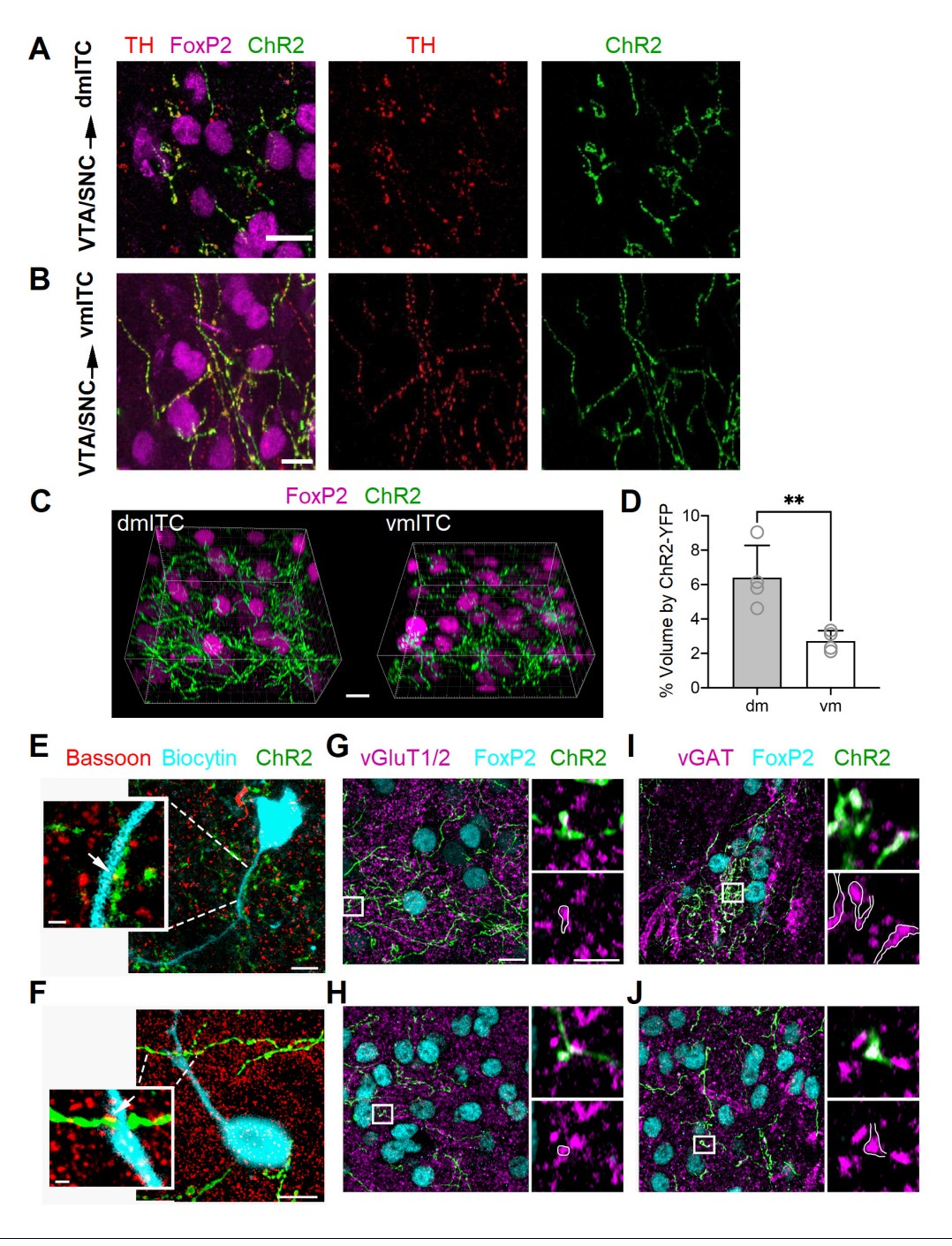

**Figure 1.** Dopaminergic fibers from VTA/SNC targeting dm- and vm-ITC clusters in the amygdala co-label with vGAT and vGluT1/2. (**A–B**) Maximum intensity projection confocal images illustrating ChR2-YFP+ axons (green) originating from tyrosine hydroxylase (TH)-positive dopaminergic neurons (TH, red) in the ventral tegmental area/ substantia nigra pars compacta (VTA/SNC) targeting FoxP2-positive neurons (magenta) in the dorsomedial (dm)- and ventromedial (vm)-intercalated cell (ITC) clusters. Overlay (left), TH staining (middle), ChR2-YFP (right). Scale bars: 10 μm. (**C**) Representative image volumes of the dm- and vm-ITC clusters with FoxP2-labeled nuclei (magenta) and ChR2-YFP+ axons (green). Scale bar: 10 μm. (**D**) The volume fraction encompassed by ChR2-YFP + axons was significantly higher in the dm-ITC (6.40 ± 0.94% of total image volume) compared to the vm-ITC cluster (2.72 ± 0.30% of total image volume; unpaired t-test, **p=0.0097). Histograms display the mean ± SEM and values from individual mice (empty circles, n=4 animals). The average dm- and vm-ITC cluster volume analyzed per animal was 138.15 ± 29.51 and 156.12 ± 10.29 μm$^3$, respectively. (**E–F**) Overlay confocal images of ChR2-

*Figure 1 continued on next page*

*Figure 1 continued*

YFP+ axons (green) and biocytin-filled cells (turquoise) in dm-ITC and vm-ITC clusters immunostained for the presynaptic marker Bassoon to examine putative active zones (white arrows). Scale bars 5 and 1 μm (inset). (**G–H**) Confocal images of dm- and vm-ITC clusters, including FoxP2-positive neurons (turquoise) that contain ChR2-YFP + fibers (green) co-labeled for the presynaptic markers vGluT1/vGluT2 (magenta). Right panels show a higher magnification of the boxed area containing an example of a bouton co-expressing ChR2-YFP and vGluT1/vGluT2 outlined in white. (**I–J**) Confocal images of dm- and vm-ITC clusters, including FoxP2-positive neurons (turquoise) that contain ChR2-YFP fibers (green) co-labeled for the presynaptic marker vGAT (magenta). Right panels show a higher magnification of the boxed area containing an example of a bouton co-expressing ChR2-YFP and vGAT outlined in white. Scale bars for (**G–J**): 10 μm for left panels, 3 μm for right panels. Thickness of confocal z-stacks: (**A**) 11.1 μM; (**B**) 12.5 μM; (**E**) 8.06 and 2.01 μm for the cell and synapse on dendrite, respectively; (**F**) 12.2 and 2.2 μm for the cell and synapse on dendrite, respectively; (**G–J**) left panels, 8.83 μm; right panels, single plane of 0.18 μm nominal thickness.

The online version of this article includes the following source data and figure supplement(s) for figure 1:

**Source data 1.** Data *Figure 1D*.
**Figure supplement 1.** Specific expression of ChR2-YFP in dopaminergic midbrain neurons.
**Figure supplement 1—source data 1.** Data *Figure 1—figure supplement 1C*.
**Figure supplement 2.** Labeling of TH-positive axons in the amygdala.
**Figure supplement 3.** Example of 3D analysis of axonal volumes in ITC clusters.
**Figure supplement 4.** Quantitative analysis TH-staining reveals stronger labeling in dm- versus vm-ITC clusters.
**Figure supplement 4—source data 1.** Data *Figure 1—figure supplement 4C*.
**Figure supplement 5.** Dm-ITC and vm-ITC clusters receive differential inputs from VTA and SNC.

receptors involved using pharmacological blockers of GABA$_A$ receptors, as well as AMPA- and NMDA-type glutamate receptors, in a subset of the recorded ITCs (*Figure 2D,G*). Applying the same approach to CeA neurons, in accordance with previous reports, we detected glutamatergic PSCs (*Mingote et al., 2015*), but also GABAergic and mixed PSCs (*Figure 2C* and *Figure 2—figure supplement 2A–B*). Overall, the relative contribution of neurotransmitters mediating the fast PSCs significantly differed between neurons in dm-ITC or vm-ITC clusters compared with CeA neurons (*Figure 2C*), with the latter having a higher proportion of glutamatergic PSCs. When retrieving locations for all recorded neurons in dm-ITC and vm-ITC clusters, and the CeA, we found that they are intermingled, with no obvious bias for a distribution along the rostro-caudal or dorso-medial axis by co-release type (*Figure 2—figure supplement 3A–B*).

In line with a monosynaptic innervation, all components of inhibitory and excitatory PSCs had short latencies in both ITC clusters and CeA (*Figure 2F and I*, *Figure 2—figure supplement 2C*). In dm-ITCs, we also directly confirmed that PSCs were monosynaptic, by demonstrating that they were abolished by the sodium channel blocker tetrodotoxin (TTX) and recovered in the presence of TTX and the potassium channel blocker 4-aminopyridine (4-AP) (*Figure 2—figure supplement 4A–B*; *Petreanu et al., 2009*). Taken together, our results strongly suggest that projections originating from VTA and SNC dopaminergic neurons mainly co-release GABA in medial ITC clusters to induce GABA$_A$ receptor-mediated fast inhibitory PSCs (IPSCs).

## Direct DA application or phasic DA afferent stimulation hyperpolarizes a fraction of dm-ITCs

A previously reported mechanism by which DA tunes ITC output and activity in younger animals is a direct DRD1-mediated hyperpolarization (*Mańko et al., 2011*; *Marowsky et al., 2005*), which has not been observed in adult animals (*Kwon et al., 2015*). Therefore, we scrutinized a direct postsynaptic action of DA using two complementary approaches: to ascertain an effect in adult animals, we first opted for DA bath application (*Figure 3A,D*), and subsequently aimed to evoke endogenous DA release by phasic stimulation of midbrain afferents (*Figure 3G*). Bath application of DA in GAD67-GFP mice induced a significant hyperpolarization in 50 and 58% of dm-ITCs from young and adult animals, respectively, as revealed by z-score analysis (*Figure 3B,E*). The hyperpolarization in responsive neurons was small (approximately −3 mV) in both age groups (*Figure 3C,F*).

Phasic activity of midbrain dopaminergic neurons has been shown to release DA in dorsal striatum and nucleus accumbens both ex vivo and in vivo (*Schultz et al., 1997*; *Stuber et al., 2010*). If

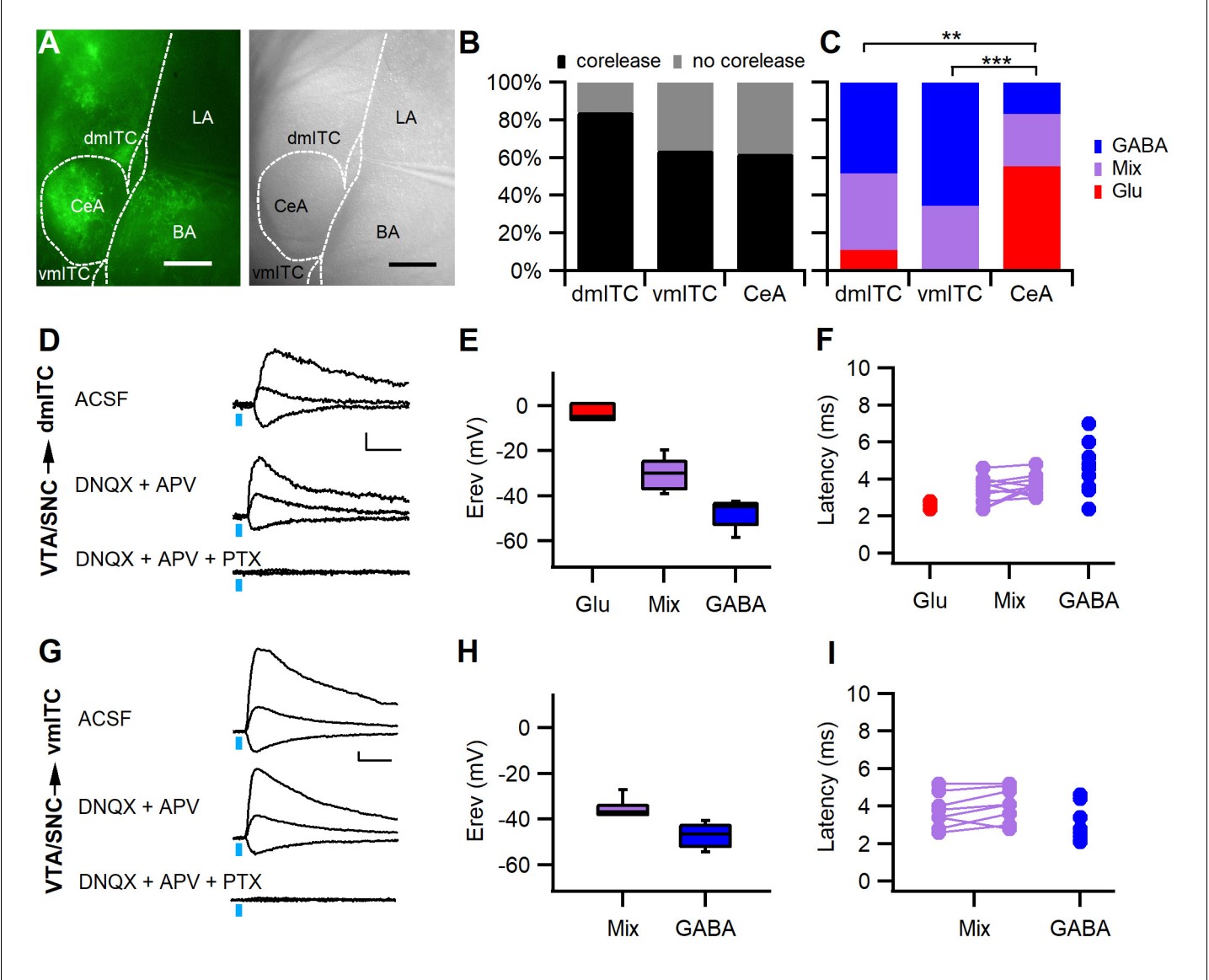

**Figure 2.** Stimulation of midbrain dopaminergic fibers evokes mainly GABAergic PSCs in dm- and vm-ITCs. (**A**) Fluorescence (left) and infrared differential interference contrast image of an amygdala slice with patch recording pipette in the dorsomedial-intercalated cell (dm-ITC) cluster (right). ChR2-YFP-expressing axons from dopaminergic midbrain were observed in central amygdala (CeA), amygdalostriatal transition zone, basal amygdala (BA), and ITC clusters. (**B**) Fast postsynaptic currents (PSCs) were evoked by stimulation of dopaminergic fibers in a large fraction of recorded ITCs and CeA neurons (Fisher's exact test=4.823, p=0.098; 84% for dm-ITCs, n=32 cells from 16 animals; 64% for ventromedial-intercalated cells (vm-ITCs), n=36 cells from 11 animals; and 62% for CeA, n=29 cells from 11 animals). (**C**) Distribution of fast PSCs in dm-ITCs, vm-ITCs, and CeA neurons by response type. ITCs show mostly GABAergic PSCs (dm-ITCs, n=27: GABA 48%, mixed 41%, Glu 11%; vm-ITCs, n=23: GABA 65%, mixed 35%), whereas CeA neurons (n=18) exhibited mostly glutamatergic PSCs (GABA 17%, mixed 28%, Glu 55%). Response types were significantly different between all regions (Fisher's exact test=21.41, p<0.001). A pairwise comparison revealed differences between CeA and ITC clusters (CeA vs. dm-ITC, **p=0.005; CeA vs. vm-ITC, ***p<0.001). (**D and G**) Representative traces of light-evoked mixed PSCs recorded at −70, 0, and 40 mV from a dm-ITC (**D**) and a vm-ITC (**G**). Application of glutamate receptor blockers DNQX (20 µM) and APV (100 µM) had a small effect, whereas the GABA$_A$-R channel blocker picrotoxin (PTX, 100 µM) abolished the PSCs entirely (n=5 cells from five animals for dm-ITC, n=3 cells from three animals for vm-ITC). Scale bars: 10 pA, 10 ms. (**E and H**) Box plot of PSC reversal potentials (Erev) for dm-ITCs (**E**) and vm-ITCs (**H**) by PSC type. (**E**) Erev in dm-ITCs was −3.49 ± 2.21 mV for glutamatergic PSCs, −48.15 ± 1.65 mV for GABAergic PSCs, and −30.21 ± 2.07 mV for mixed PSCs (one-way ANOVA, F(2, 24)=70.563, p<0.001). (**H**) Erev in vm-ITCs was −47.12 ± 1.31 mV for GABAergic PSCs and −35.57 ± 1.32 mV for mixed PSCs (unpaired t-test, p<0.001). (**F and I**) Latencies of PSCs were consistent with monosynaptic connections. (**F**) Latencies in dm-ITCs were 2.60 ± 0.12 ms (n=3) for pure glutamatergic, 3.45 ± 0.21 ms for the glutamatergic, and 3.67 ± 0.16 ms (n=11) for the GABAergic components of mixed PSCs, and 4.33 ± 0.35 ms (n=13) for pure GABAergic PSCs. (**I**) Latencies in vm-ITCs were 3.75 ± 0.32 ms for glutamatergic and 4.09 ± 0.32 ms (n=8) for GABAergic components of mixed PSCs, and 3.04 ± 0.25 ms (n=15) for pure GABAergic PSCs.

*Figure 2 continued on next page*

*Figure 2 continued*

The online version of this article includes the following source data and figure supplement(s) for figure 2:

**Source data 1.** Data *Figure 2C, E-F*.
**Source data 2.** Data *Figure 2C, H-I*.
**Source data 3.** Data *Figure 2C* and *Figure 2—figure supplement 2B-C*.
**Figure supplement 1.** Overview of ChR2-YFP injection and expression sites in the midbrain for co-release experiments.
**Figure supplement 2.** Fast PSCs onto CeA neurons are mainly glutamatergic.
**Figure supplement 3.** Locations of neurons receiving co-release and no co-release from DA midbrain afferents.
**Figure supplement 4.** Fast PSCs from midbrain dopaminergic fibers onto dm-ITCs are monosynaptic.
**Figure supplement 4—source data 1.** Data *Figure 2—figure supplement 4B*.

this is also the case for ITCs, we should be able to detect an electrophysiological signature similar to that of direct DA application. To test this, we optogenetically activated DA midbrain afferents in adult DAT-Cre mice with 30 Hz phasic stimulation, and monitored membrane potential changes in the presence of GABA$_A$- and glutamate-R blockers, to exclude postsynaptic effects of co-released fast neurotransmitters (*Figure 3G*). In keeping with results from direct DA application, we found a significant hyperpolarization upon activation of dopaminergic fibers in 69% of dm-ITCs (*Figure 3H,I*) that persisted in the presence of the GABA$_B$-R blocker CGP55845 (*Figure 3—figure supplement 1*). Importantly, the magnitude of the hyperpolarization was similar to that of DA application (*Figure 3C,F,I*), suggesting that bath application of DA is a good proxy for physiologically released DA. In all of the experiments, we found no obvious bias for a specific rostro-caudal or dorso-medial location of DA-responsive neurons (*Figure 3—figure supplement 2A–C*). In conclusion, our data point to the fact that DA is released upon phasic stimulation of midbrain afferents and induces a small hyperpolarization in a fraction of dm-ITCs.

## Phasic stimulation of dopaminergic fibers alters sIPSC amplitude in ITCs

Apart from controlling cellular excitability, DA can also modulate excitatory and inhibitory synaptic transmission by pre- and postsynaptic mechanisms (*Tritsch et al., 2012*). We next explored the effect of endogenously released DA on ITC synaptic inputs and within ITC cluster inhibitory interactions. Toward this end, we stimulated dopaminergic midbrain afferents from confirmed midbrain injection sites (*Figure 4—figure supplement 1*) with 30 Hz light pulse trains and monitored the frequency and amplitude of spontaneous synaptic currents onto dm-ITCs before and after stimulation (*Figure 4A*). While we did not detect changes in the frequency, we observed a small but significant reduction in the amplitude of spontaneous inhibitory postsynaptic currents (sIPSCs) (*Figure 4A–B*). In keeping with a role of DA in modulating sIPSCs, the effect of 30 Hz stimulation was completely blocked in the presence of a cocktail of DA-R blockers (*Figure 4C–D*). Conversely, tonic 1 Hz stimulation of DA midbrain afferents did not affect frequency or amplitude of sIPSCs (*Figure 4E–F*). Indeed, direct comparison of sIPSC amplitude modulation upon phasic 30 Hz stimulation in the absence or presence of DA-R blockers revealed significant differences (*Figure 4G*), as did a direct comparison between phasic and tonic stimulation (*Figure 4H*). Interestingly, the observed changes were specific for sIPSCs, as we did not detect any changes in the frequency or amplitude of spontaneous excitatory postsynaptic currents (sEPSCs) (*Figure 4—figure supplement 2A–B*). Stimulation of dopaminergic midbrain inputs at 30 Hz also decreased the amplitude of sIPSC in vm-ITCs (*Figure 4—figure supplement 3A–B*), corroborating our findings from dm-ITCs. In summary, our results indicate that phasic release of DA has a neuromodulatory effect on inhibitory inputs in dm- and vm-ITC networks. This could stem from either reduced excitability in a fraction of ITCs in the network or a direct modulatory action at inhibitory synapses.

## DA modulates inhibitory synaptic transmission between ITC clusters

Following our observation that phasic stimulation of midbrain dopaminergic afferents affects sIPSCs, we wanted to more directly dissect the effect of DA on defined inhibitory synapses. Local inhibitory interactions have been shown between ITCs within one cluster, and also between medially located ITC clusters (*Busti et al., 2011*; *Geracitano et al., 2007*; *Royer et al., 2000*). To examine the effect of DA on between-cluster synaptic interactions, we used FoxP2-Cre mice to express AAV-ChR2-YFP

specifically and selectively in one of the ITC clusters. In subsequent slice recordings, we stimulated axons and recorded in the innervated target cluster (*Figure 5A*, *Figure 5—figure supplement 1A, D*). For the dm-ITCs, we hypothesized that the lateral ITC (l-ITC) cluster provides one of the inputs.

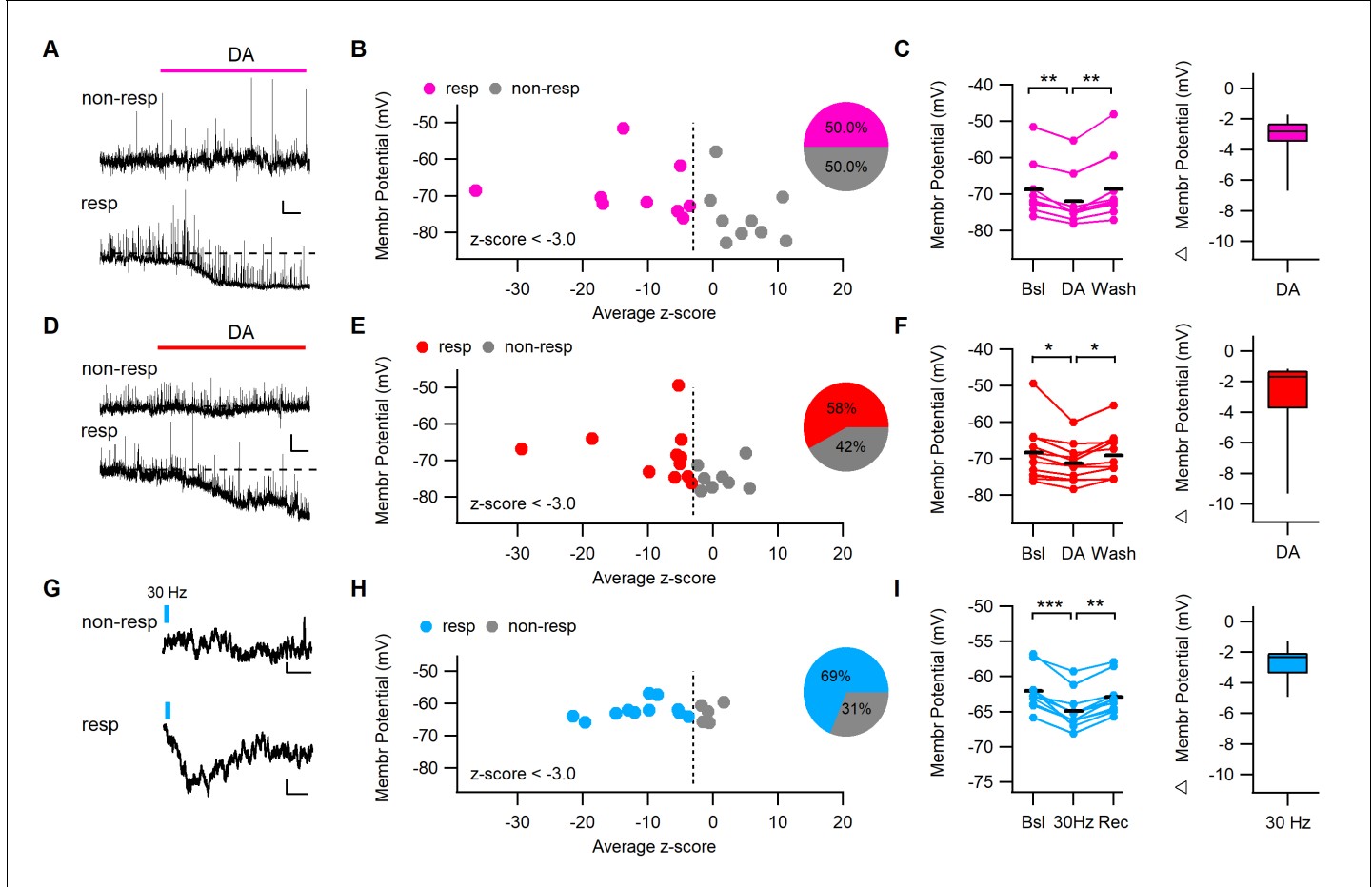

**Figure 3.** DA application or phasic stimulation of midbrain inputs hyperpolarizes a fraction of dm-ITCs. (**A, D**) Representative traces of dorsomedial-intercalated cells (dm-ITCs) recorded at resting membrane potential in current clamp mode from young (postnatal day 20–28) and adult (8–10 weeks) mice. Dopamine (DA, 15–30 µM) was applied during the time indicated. Scale bars for (A): 3 mV, 30 s; and for D: 2 mV, 30 s. DA hyperpolarized some cells (responding), but not others (non-responding). (**B, E**) Distribution of recorded cells according to how DA affects their membrane potential represented as z-scores (x-axis) and their initial membrane potential (y-axis). Responding cells (z-score cut-off at −3) are shown in pink (young) or red (adult), non-responding cells in grey. Insets: fraction of DA-responsive neurons in young mice (n=18 cells from seven animals) and adult mice (n=19 cells from 10 animals). (**C, F**) Left: Absolute and relative changes in membrane potential in DA-responsive neurons recorded from young (**C**) and adult mice (**F**). DA significantly hyperpolarized responsive dm-ITCs in slices from young (n=9 cells from five animals, repeated-measures ANOVA: $F_{(2, 16)}$=16.23, p<0.001; Bonferroni post-hoc test: Bsl vs. DA **p=0.001, DA vs. Wash **p=0.004) and adult animals (n=11 cells from eight animals, repeated-measures ANOVA: $F_{(2, 20)}$=7.91, p=0.003; Bonferroni post-hoc test: Bsl vs. DA *p=0.016, DA vs. Wash *p=0.019). DA-induced hyperpolarization amounted to −3.18 ± 0.48 mV and −2.94 ± 0.83 mV in neurons from young and adult mice, respectively. (**G**) Representative traces of dm-ITCs recorded in current clamp mode from adult mice. Trains of 10 pulses at 30 Hz optogenetic stimulation of DA midbrain afferents were applied during the time indicated. Scale bars: 0.5 mV, 1 s. (**H**) Distribution of recorded cells according to 30 Hz stimulation of DA afferents affects their membrane potential represented as z-scores (x-axis) and their initial membrane potential (y-axis). Responding cells (z-score cut-off at −3) are shown in blue, non-responding cells in grey. Insets: Fraction of stimulation-responsive neurons (n=16 cells from three animals). (**I**) Absolute and relative changes in membrane potential in responsive neurons. 30 Hz stimulation significantly hyperpolarized responsive dm-ITCs (n=11 cells from two animals, repeated-measures ANOVA: $F_{(2, 20)}$=49.01, p<0.001; Bonferroni post-hoc test: Bsl vs. Stim *p<0.001, Stim vs. Recovery *p=0.001). 30 Hz stimulation-induced hyperpolarization amounted to −2.84 ± 0.35 mV.

The online version of this article includes the following source data and figure supplement(s) for figure 3:

Source data 1. Data *Figure 3B, C, E, F and H-I*, and *Figure 3—figure supplement 1*.

Figure supplement 1. 30 Hz stimulation hyperpolarizes dm-ITCs in the presence of a GABA_B receptor blocker.

Figure supplement 2. Distribution of dm-ITCs according to their response to DA and 30 Hz stimulation.

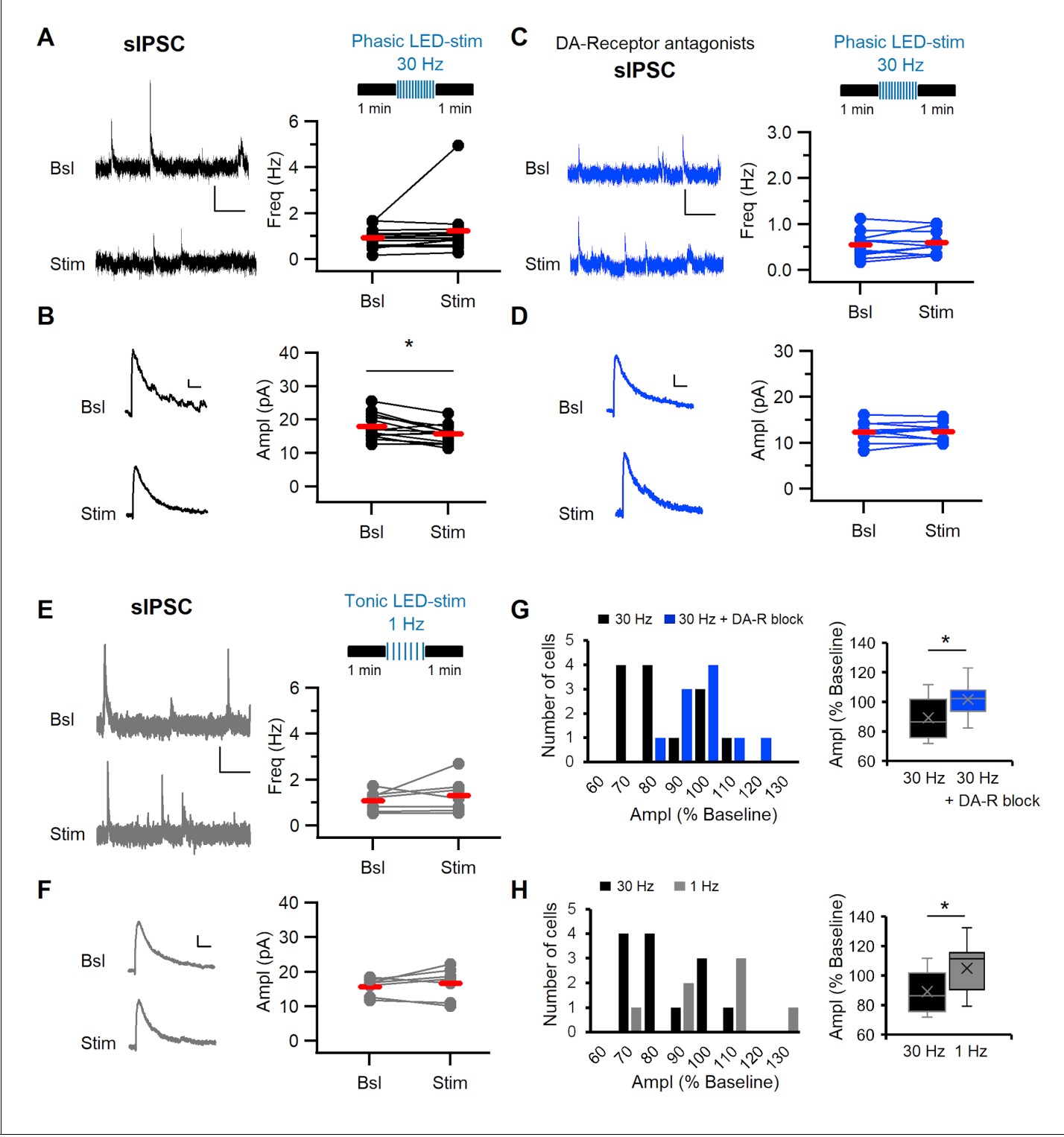

**Figure 4.** Phasic stimulation of midbrain dopaminergic fibers alters sIPSC amplitude in dm-ITCs in a DA-R-dependent manner. (A, C, E) Top: Experimental protocol. Spontaneous inhibitory postsynaptic current (sIPSC) frequency and amplitude were assessed before and after optogenetic stimulation of midbrain dopaminergic fibers in (A, C) with phasic (10 times 10 pulses at 30 Hz) and in (E) with tonic (100 pulses at 1 Hz) patterns. (A–F) Left: sIPSC example traces before (Bsl) and after stimulation (Stim). All sIPSCs were recorded at 0 mV. Right: Plots of sIPSC frequency and amplitude depict values for individual neurons (dots) and the average (red line). (A) Left: Example traces of sIPSC activity (scale bars: 10 pA, 1 s). Right: sIPSC frequency did not change upon 30 Hz stimulation (0.94 ± 0.12 Hz vs. 1.23 ± 0.32 Hz, n=13, paired t-test, p=0.278). (B) Left: Example traces of averaged sIPSCs (scale bars: 2 pA, 20 ms). Right: sIPSC amplitude decreased after 30 Hz stimulation (17.94 ± 1.06 pA vs. 15.74 ± 0.78 pA, n=13, paired t-test,

*Figure 4 continued on next page*

*Figure 4 continued*

*p=0.011). (C) Left: Example traces of sIPSC activity (scale bars 10 pA, 1 s). Right: sIPSC frequency did not change (0.55 ± 0.09 Hz vs. 0.60 ± 0.08 Hz, n=10, paired t-test, p=0.349) in the presence of DA-R blockers (D1: SCH23390 10 μM; D2: Sulpiride 20 μM; D4: L-745870 100 nM). (D) Left: Example traces of averaged sIPSCs (scale bars 2 pA, 20 ms). sIPSC amplitude did not change after 30 Hz stimulation in the presence of DA-R blockers (12.36 ± 0.72 pA vs. 12.43 ± 0.64 pA, n=10, paired t-test, p=0.857). (E) Left: Example traces of sIPSC activity (scale bars 10 pA, 1 s). sIPSC frequency did not change after 1 Hz stimulation (1.07 ± 0.17 Hz vs. 1.31 ± 0.28 Hz, n=7, paired t-test, p=0.348). (F) Left: Example traces of averaged sIPSCs (scale bars 2 pA, 20 ms). Right: sIPSC amplitude did not change after 1 Hz stimulation (15.71 ± 0.96 pA vs. 16.68 ± 1.74 pA, n=7, paired t-test, p=0.398). (G) Comparison of amplitude changes in sIPSCs after 30 Hz stimulation in the absence (black) and in the presence of DA-R blockers (blue). Left: Distribution of amplitudes (as % of baseline). Right: Box plots comparing relative amplitudes between the groups (Mann-Whitney U-test, *p=0.047), indicating that DA-R blockers prevent the effect of 30 Hz stimulation on sIPSCs. (H) Comparison of amplitude changes in sIPSCs after 30 Hz (black) and 1 Hz stimulation (grey) of dopaminergic fibers. Left: Distribution of amplitudes (as % of baseline). Right: Box plots comparing relative amplitudes between the groups (Mann-Whitney U-test, *p=0.043). Data were obtained from three to seven animals per experiment.

The online version of this article includes the following source data and figure supplement(s) for figure 4:

**Source data 1.** Data *Figure 4A-F*.
**Figure supplement 1.** Overview of ChR2-YFP injection and expression sites in the midbrain for phasic and tonic stimulation experiments.
**Figure supplement 2.** Phasic stimulation of midbrain dopaminergic fibers does not affect sEPSCs in dm-ITCs.
**Figure supplement 2—source data 1.** Data *Figure 4—figure supplement 2A–B*.
**Figure supplement 3.** Phasic stimulation of midbrain dopaminergic fibers alters sIPSC amplitude in vm-ITCs.
**Figure supplement 3—source data 1.** Data *Figure 4—figure supplement 3A–B*.

Indeed, optical stimulation of ChR2-YFP+ fibers arising from l-ITCs reliably evoked PSCs in dm-ITCs (*Figure 5—figure supplement 1B*), which reversed around the equilibrium potential for chloride (Erev = −44.77 ± 1.77, n=6 cells from three animals), and was largely abolished by the GABA$_A$-channel blocker picrotoxin (PTX, % remaining current at 0 mV = 4.70 ± 1.09%, n=3 cells from two animals). This suggests that l-ITCs provide fast GABAergic inputs to the dm-ITC cluster. Next, we assessed the effect of DA on inhibitory transmission in l-ITC→dm-ITC and dm-ITC→vm-ITC pathways. Optically evoked IPSCs in the l-ITC→dm-ITC and dm-ITC→vm-ITC pathways were both of short latency, in keeping with monosynaptic connectivity (*Figure 5—figure supplement 1C*). To select for effects of dopaminergic modulation of IPSCs without interference by GABA co-release from dopaminergic fibers, we opted to bath apply DA for 5 min. To gain insight into a possible presynaptic action, we used a paired pulse stimulation protocol (100 ms interstimulus interval). DA application significantly and reversibly suppressed IPSC amplitude to approximately 50% of baseline in l-ITC→dm-ITC (*Figure 5—figure supplement 1E,F*) and dm-ITC→vm-ITC pathways (*Figure 5B–C*), which was accompanied by a significant increase in the paired pulse ratio (PPR) (*Figure 5—figure supplement 1F*, *Figure 5C*). Taken together, our data are consistent with a presynaptic site of action and indicate that DA dampens inhibitory interactions between l-ITC→dm-ITC and dm-ITC→vm-ITC clusters by decreasing GABA release. This is further supported by the fact that the decrease in IPSC amplitude was significantly correlated with the associated increase in PPR (*Figure 5J*).

## Presynaptic DRD1 activation mimics the effect of DA in suppressing inhibitory interactions between ITC clusters

Dopamine receptors DRD1 and DRD2 have been localized in the amygdala, with ITC clusters being particularly enriched in DRD1 (*Fuxe et al., 2003*; *Pinto and Sesack, 2008*). To identify which receptor is involved in the DA-induced suppression of inhibitory synaptic transmission between ITC clusters, we used dihydrexidine (DH) and quinpirole as selective agonists for DRD1 and DRD2, respectively.

Activation of DRD1 significantly suppressed IPSCs similar to DA (to about 50%) in the dm-ITC→vm-ITC pathway, whereas DRD2 activation had no significant effect on amplitude and was different from that of DA or DRD1 activation (*Figure 5D–E*). Activation of DRD1 also significantly increased the PPR in the dm-ITC→vm-ITC pathway, but DRD2 activation did not (*Figure 5D,F*). The DH-induced IPSC suppression and PPR increase were comparable to the effect of DA application (*Figure 5E–F*). Likewise, DRD1 activation also suppressed IPSCs in the l-ITC→dm-ITC pathway, whereas DRD2 activation had only a small, yet significant, effect on amplitude, which was different from that of DA or DRD1 activation (*Figure 5G–H*). Furthermore, activation of DRD1 significantly increased the PPR in the l-ITC→dm-ITC pathway, whereas DRD2 activation did not (*Figure 5G,I*).

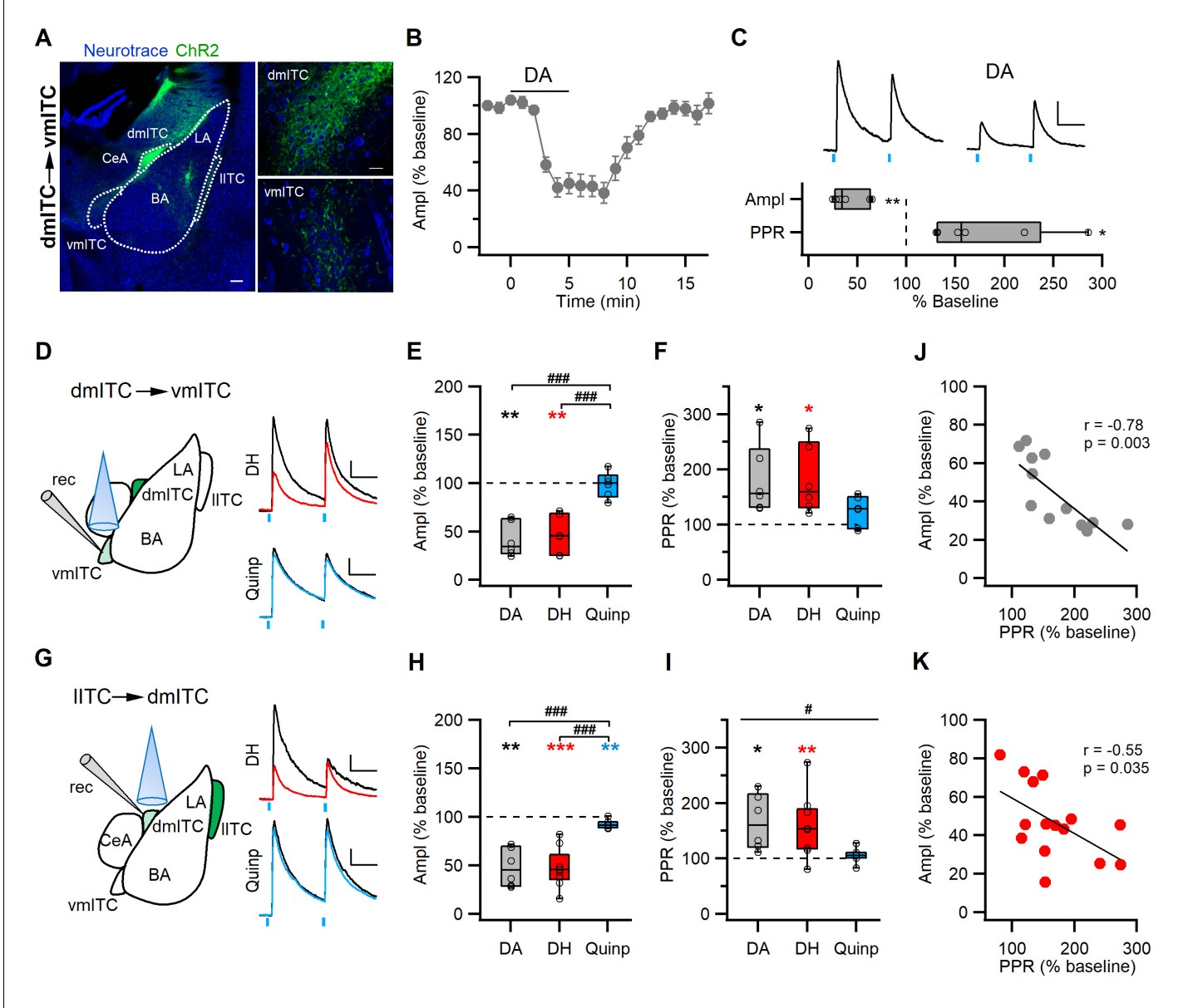

**Figure 5.** DA via DRD1 mediates presynaptic depression between ITC clusters. (**A**) Confocal images of the amygdala showing ChR2-YFP-expressing neurons in the dorsomedial-intercalated cell (dm-ITC) cluster and their efferents targeting the ventromedial-intercalated cell (vm-ITC) cluster after injection of AAV-DIO-ChR2-YFP in FoxP2-Cre mice. Left: Overview of the amygdala. Right: Details of the dm-ITC injection site with YFP+ cells (top) and vm-ITC recording site with YFP+ fibers (bottom). Scale bars: 200 and 20 μm. (**B**) Time course of changes in inhibitory postsynaptic current (IPSC) amplitude upon bath application of dopamine (DA, 30 μM, 5 min) in the dm-ITC→vm-ITC pathway. DA decreased IPSC amplitude, which returned to near baseline levels upon washout. (**C**) Top: Representative IPSC traces recorded at 0 mV from the baseline period and during DA application (paired pulse interval 100 ms). Scale bars: 50 pA, 50 ms. Bottom: Relative change of IPSC amplitude and paired pulse ratio (PPR) during DA application (n=6 cells from three animals, amplitude 41.53 ± 7.25%, paired t-test, **p=0.003; PPR 180.00 ± 25.01%, paired t-test, *p=0.031). (**D and G**) Left: Schematics of experimental approach with infection, recording, and light stimulation sites. Right: Representative traces recorded at baseline (black) and during agonist application (colored). Dihydrexidine (DH, 10 μM), a DRD1 agonist, is shown in red, Quinpirole (Quinp, 1 μM), a DRD2 agonist, is shown in blue (paired pulse interval: 100 ms). Scale bars 50 pA, 50 ms. (**E and H**) Comparison of relative changes in IPSC amplitude during agonist application in the dm-ITC→vm-ITC pathway (**E**) and l-ITC→dm-ITC pathway (**H**). (**E**) DH (red, n=6 cells from four animals) suppressed IPSCs (amplitude 46.69 ± 8.14%, **p=0.010, paired t-test), whereas Quinp (blue, n=6 cells from three animals) had no effect (amplitude 98.45 ± 5.37, p=0.438, paired t-test). Between-group analysis revealed a significant drug effect on the dm-ITC→vm-ITC amplitude (one-way ANOVA, F(2, 15)=20.107, p<0.001) with Quinp differing from DA and DH (###p<0.001 each, Bonferroni post-hoc tests). (**H**) DH (red, n=9 cells from five animals) strongly suppressed IPSCs (amplitude 47.16 ± 6.67%, ***p<0.001, paired t-test), whereas Quinp (blue, n=7 cells from three animals) had a minor effect (amplitude 92.38 ± 1.63%, **p=0.006). Between-group analysis revealed a significant drug effect on the l-ITC→dm-ITC amplitude (one-way ANOVA, F(2, 19)=17.394, p<0.001) with Quinp differing from

*Figure 5 continued on next page*

*Figure 5 continued*

DA and DH (###p<0.001 each, Bonferroni post-hoc tests). (**F and I**) Comparison of relative change of PPR during drug application in the dm-ITC→vm-ITC pathway and l-ITC→dm-ITC pathway. (**F**) DH (red, n=6) increased PPR (181.37 ± 25.38%, *p=0.035, paired t-test), whereas Quinp (blue, n=6) had no significant effect on PPR (123.97 ± 11.25%, p=0.127, paired t-test). Between-group analysis revealed no significant drug effect on the dm-ITC→vm-ITC PPR (one-way ANOVA, F(2, 15)=2.304, p=0.134). (**I**) DH (red, n=9) increased PPR (158.46 ± 18.52%, **p=0.005, paired t-test), whereas Quinp (blue, n=7) had no effect on PPR (105.02 ± 4.98%, p=0.385, paired t-test). Between-group analysis revealed a significant drug effect on the l-ITC→dm-ITC PPR (one-way ANOVA, F(2, 19)=3.804, #p=0.041). (**J and K**) Combined data from both pathways show a significant correlation between change in PPR and change in amplitude upon application of DA (J: Pearson correlation, r=−0.78, p=0.003, n=12) and upon application of DH (K: Pearson correlation, r=−0.55, p=0.035, n=15).

The online version of this article includes the following source data and figure supplement(s) for figure 5:

**Source data 1.** Data *Figure 5B*.
**Source data 2.** Data *Figure 5C*.
**Source data 3.** Data *Figure 5E-F*.
**Source data 4.** Data *Figure 5H–I*.
**Figure supplement 1.** An l-ITC→dm-ITC inhibitory pathway that is modulated by DA via a presynaptic mechanism.
**Figure supplement 1—source data 1.** Data *Figure 5—figure supplement 1C*.
**Figure supplement 1—source data 2.** Data *Figure 5—figure supplement 1E*.
**Figure supplement 1—source data 3.** Data *Figure 5—figure supplement 1F*.

Importantly, we again observed a significant correlation between the decrease in IPSC amplitude and the increase in PPR upon application of the DRD1 agonist (*Figure 5K*), supporting a presynaptic site of action for DRD1.

## Early extinction training alters midbrain input properties and modulation of ITCs

Recent data show that activity in midbrain dopaminergic neurons correlates strongly with early extinction learning, suggesting that, in the amygdala, DA provides a prediction error-like neuronal signal that is necessary to initiate fear extinction (*Cai et al., 2020*; *Luo et al., 2018*; *Salinas-Hernández et al., 2018*). Thus, we tested if early extinction training alters co-release from midbrain dopaminergic inputs to ITCs and the dopaminergic modulation of ITC cluster interaction. To this end, we subjected one group of mice to fear conditioning on day 1 and a day later to fear retrieval and a partial extinction training (early extinction, E-Ext) by presenting 16 CSs in the absence of the reinforcing US, while the control group was only exposed to the same number of CSs (CS-only, *Figure 6A*). We deliberately used this training protocol that does not induce a significant reduction in freezing levels, yet, to capture a time point for ex vivo investigations when the prediction error that drives learning should be large due to the US omission. As expected, in both DAT-Cre and FoxP2-Cre mice transduced with ChR2-YFP in midbrain and ITCs, respectively, this short extinction protocol did not reduce fear expression significantly (*Figure 6—figure supplement 1A–B* and *Figure 6—figure supplement 3A–B*). Importantly, these mice can extinguish when subjected to more CSs (up to 25) on day 2 and an additional day of extinction training (*Figure 6—figure supplement 2*). As our focus was on early extinction, we opted to obtain brain slices for recordings only after early extinction training (*Figure 6A*).

Early extinction training significantly decreased the reversal potential of fast PSCs evoked by stimulation of dopaminergic inputs from confirmed midbrain injection sites (*Figure 6—figure supplement 1C*) onto dm-ITCs compared to animals in the CS-only group (*Figure 6B–C*). When categorizing PSCs according to their reversal potential, as described for naive animals (cf. *Figure 2C–F*), we found that the pattern of responses shifted to more GABAergic PSCs with early extinction training (*Figure 6D*). This suggests that DA midbrain inputs are more likely to co-release GABA, which can contribute to inhibit dm-ITCs. Secondly, we found that early extinction training altered the efficacy of dopaminergic modulation of ITC cluster interaction in the dm-ITC→vm-ITC pathway (*Figure 6E–G*). The suppression of IPSC amplitude by DA was significantly larger in neurons recorded from animals in the E-Ext compared to the CS-only group, with a concomitant significantly larger increase in the PPR of IPSCs (*Figure 6F–G*). This suggests that early extinction training decreases inhibition by enhancing a dopaminergic presynaptic mechanism, which can contribute to the disinhibition of vm-ITCs. In summary, our data indicate that changes in dopaminergic input

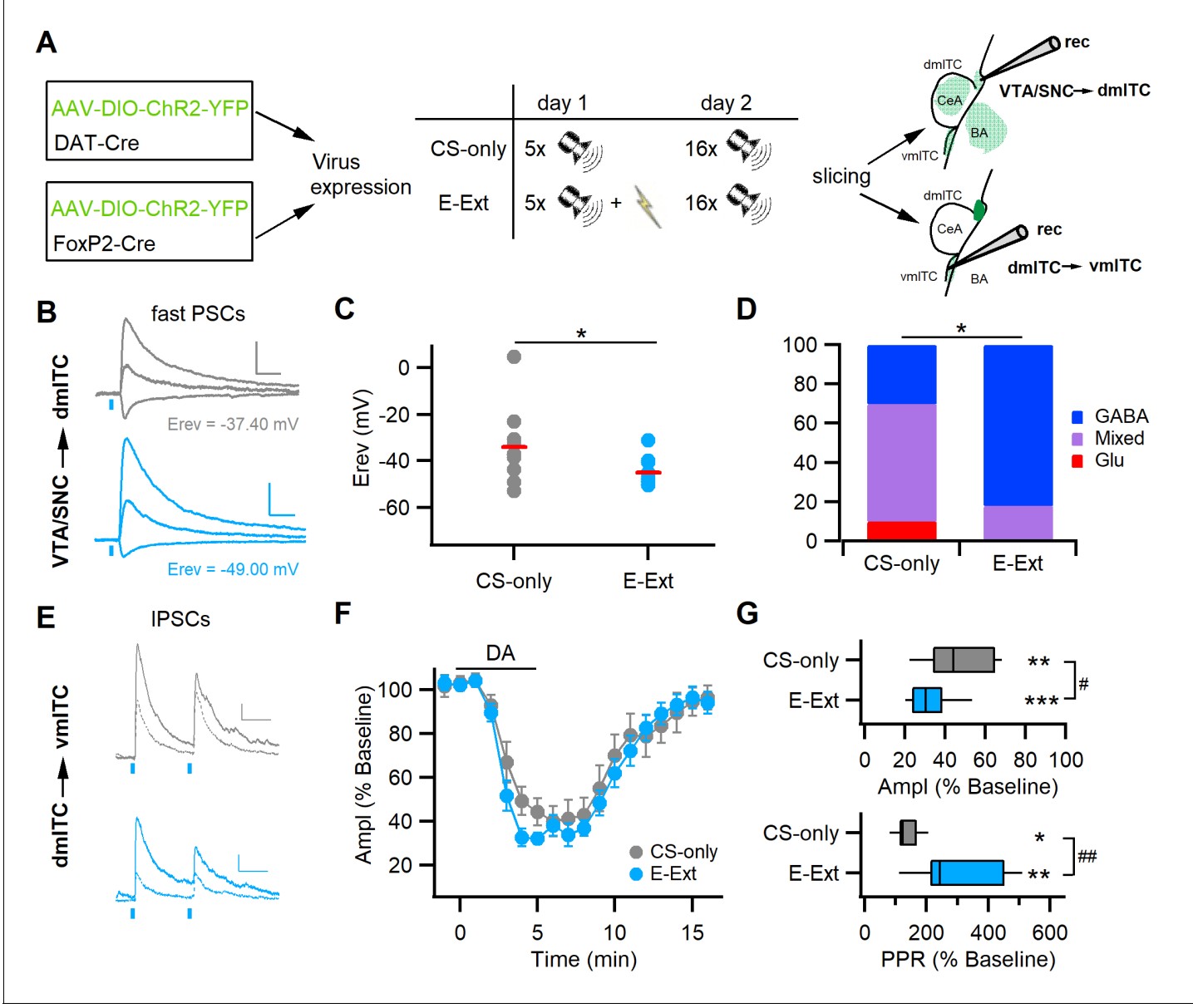

**Figure 6.** Early extinction enhances GABA release from midbrain terminals and DA-mediated depression of the dm-ITC→vm-ITC pathway. (**A**) Experimental scheme: To investigate the VTA/SNC→dm-ITC pathway, ChR2 was transduced in dopaminergic midbrain neurons of DAT-Cre mice. To investigate dopamine (DA) modulation of the dm-ITC→vm-ITC pathway, ChR2 was transduced into the dorsomedial-intercalated cell (dm-ITC) cluster of FoxP2-Cre mice. The early extinction group (E-Ext) underwent fear conditioning on day 1 (5 conditioned stimulus-unconditioned stimulus (CS-US) pairings) and early extinction training on day 2 (16 CS presentations). The CS-only group received only CS presentations. (**B**) Left: Example traces of light-evoked postsynaptic currents (PSCs) by dopaminergic fiber stimulation recorded in dm-ITCs at −70, 0, and 40 mV from CS-only (grey traces) or E-Ext animals (blue traces). Scale bars: 50 pA, 50 ms. (**C**) Plot of PSC reversal potentials in individual dm-ITCs (dots) and average (red lines) from CS-only and E-Ext groups. Erev was significantly lower in the E-Ext (−45.09 ± 1.76 mV, n=11 cells from six animals) vs. the CS-only group (−34.06 ± 5.11 mV, n=10 cells from four animals, *p=0.047, paired t-test). (**D**) Summary graph comparing the type of fast PSCs in dm-ITCs recorded from CS-only and E-Ext animals (CS-only, n=10: GABA 30%, mixed 60%, Glu 10%; E-Ext, n=11: GABA 82%, mixed 18%). PSC types were significantly different between groups (Fisher's exact test = 5.68, *p=0.041). (**E**) Example traces of light-evoked inhibitory postsynaptic currents (IPSCs) recorded in a ventromedial-intercalated cell (vm-ITC) at 0 mV upon paired pulse stimulation (100 ms interstimulus interval) of the dm-ITC→vm-ITC pathway from CS-only (grey traces) or E-Ext (blue traces) animals before (solid) and during DA application (dotted). Scale bars 50 pA, 50 ms. (**F**) Time course of changes in IPSC amplitude upon bath application of DA (30 μM, 5 min) in dm-ITC→vm-ITC pathway in CS-only and E-Ext groups. Two-way ANOVA (1–9 min) revealed significant changes for time, F(8)=37.903, p<0.001, and group, F(1)=8.229, p=0.005, but no significant interaction, F(8, 144)=0.521, p=0.839. (**G**) Significant changes of IPSC amplitude (paired t-tests: CS-only, **p=0.002; E-Ext, ***p<0.001) and paired pulse ratio (PPR) (paired t-tests: CS-only, *p=0.040; E-Ext, **p=0.003) 4–5 min after DA application in both groups. IPSC amplitude was more depressed in neurons recorded from E-Ext vs. CS-only animals

*Figure 6 continued on next page*

*Figure 6 continued*

(32.42 ± 3.16%, n=11 cells from six animals, vs. 46.75 ± 6.14%, n=7 cells from four animals, unpaired t-test, #p=0.036). The PPR increase was larger in neurons recorded from E-Ext vs. CS-only animals (310.63 ± 42.64%, n=11, vs. 139.68 ± 15.43%, n=7, unpaired t-test, ##p=0.007).

The online version of this article includes the following source data and figure supplement(s) for figure 6:

**Source data 1.** Data *Figure 6C–D*.
**Source data 2.** Data *Figure 6F*.
**Source data 3.** Data *Figure 6G*.
**Figure supplement 1.** Behavioral data from DAT-Cre mice used for ex vivo recordings and viral injection sites.
**Figure supplement 1—source data 1.** Data *Figure 6—figure supplement 1B*.
**Figure supplement 2.** Behavioral data from mice trained with a long extinction protocol.
**Figure supplement 2—source data 1.** Data *Figure 6—figure supplement 2*.
**Figure supplement 3.** Behavioral data from FoxP2-Cre mice used for ex vivo recordings.
**Figure supplement 3—source data 1.** Data *Figure 6—figure supplement 3B*.

action during early extinction learning can switch the activity balance between ITC clusters by enhancing inhibition of dm-ITCs and fostering disinhibition of vm-ITCs.

## Discussion

Here, we investigated the functional impact of midbrain dopaminergic inputs and dopaminergic modulation onto amygdala ITC clusters. Our key findings are that ITCs are controlled by several distinct mechanisms. These include fast inhibition resulting from a prominent co-release of GABA and slower mechanisms by a long-lasting DA-induced hyperpolarization, as well as a DRD1-mediated presynaptic suppression of inhibitory interactions between distinct ITC clusters (*Figure 7A*). Upon early extinction learning, fast inhibition onto dm-ITCs is increased, and DA more potently suppresses dm-ITC cluster-mediated inhibition of vm-ITCs. This may support a shift in the activity balance between these two distinct ITC clusters by inhibiting dm-ITCs and disinhibiting vm-ITCs to enable fear suppression during extinction learning (*Figure 7B*).

Although it is well established that amygdala ITC clusters receive dense TH+ projections (*Asan, 1997*; *Fuxe et al., 2003*), here, we show that these are part of the mesolimbic and nigrostriatal pathways from VTA/SNC, also providing afferents to amygdalostriatal transition zone (Astria), BA, and CeA. The current knowledge about localization and molecular phenotype of amygdala-projecting midbrain neurons is still incomplete and, due to the lack of specific tools, has remained elusive for ITCs. Amygdala-projecting neurons are localized both in SNC and VTA, and a distinct population of DA neurons in DR/PAG targets CeA (*Beier et al., 2015*; *Groessl et al., 2018*; *Poulin et al., 2018*; *Vogt Weisenhorn et al., 2016*). A fraction of BLA-projecting neurons in VTA also expresses vGluT2 and/or is localized in medial VTA regions containing TH+/vGluT2+ neurons (*Morales and Margolis, 2017*; *Poulin et al., 2018*). The only evidence on the source of ITC inputs comes from a 6-OHDA lesion study (*Ferrazzo et al., 2019*) suggesting that ITCs are targeted by SNC/VTA regions that also target BLA. Our results are well in line with this and, additionally, suggest a differential targeting of dm- versus vm-ITC clusters by VTA and SNC. Consistent with previous ultrastructural investigations demonstrating that TH+ axons contact ITC somata, dendrites, and spines (*Asan, 1997*; *Pinto and Sesack, 2008*), we detected putative presynaptic terminals originating from VTA/SNC on the soma and along ITC dendrites, suggesting functional connectivity.

While co-release of glutamate and GABA from VTA/SNC dopaminergic neurons is well established in nucleus accumbens and dorsal striatum (*Granger et al., 2017*), glutamate co-release has only recently been demonstrated in CeA (*Mingote et al., 2015*). Here, we also find GABA co-release in CeA, although to a much lower extent than glutamate co-release. We did not observe glutamate co-release in the BLA, which is in line with a previous study (*Mingote et al., 2015*); however, see also *Lutas et al., 2019*.

Our study is the first to directly examine dopaminergic afferents onto amygdala ITCs via optogenetic stimulation. Combined qualitative anatomical and quantitative electrophysiological evidence strongly supports the notion that glutamate and GABA can be released from dopaminergic afferents onto ITCs, with a major contribution of GABA. GABA co-release is a prominent feature of midbrain dopaminergic neurons targeting dorsal striatum and nucleus accumbens, where it can rapidly inhibit

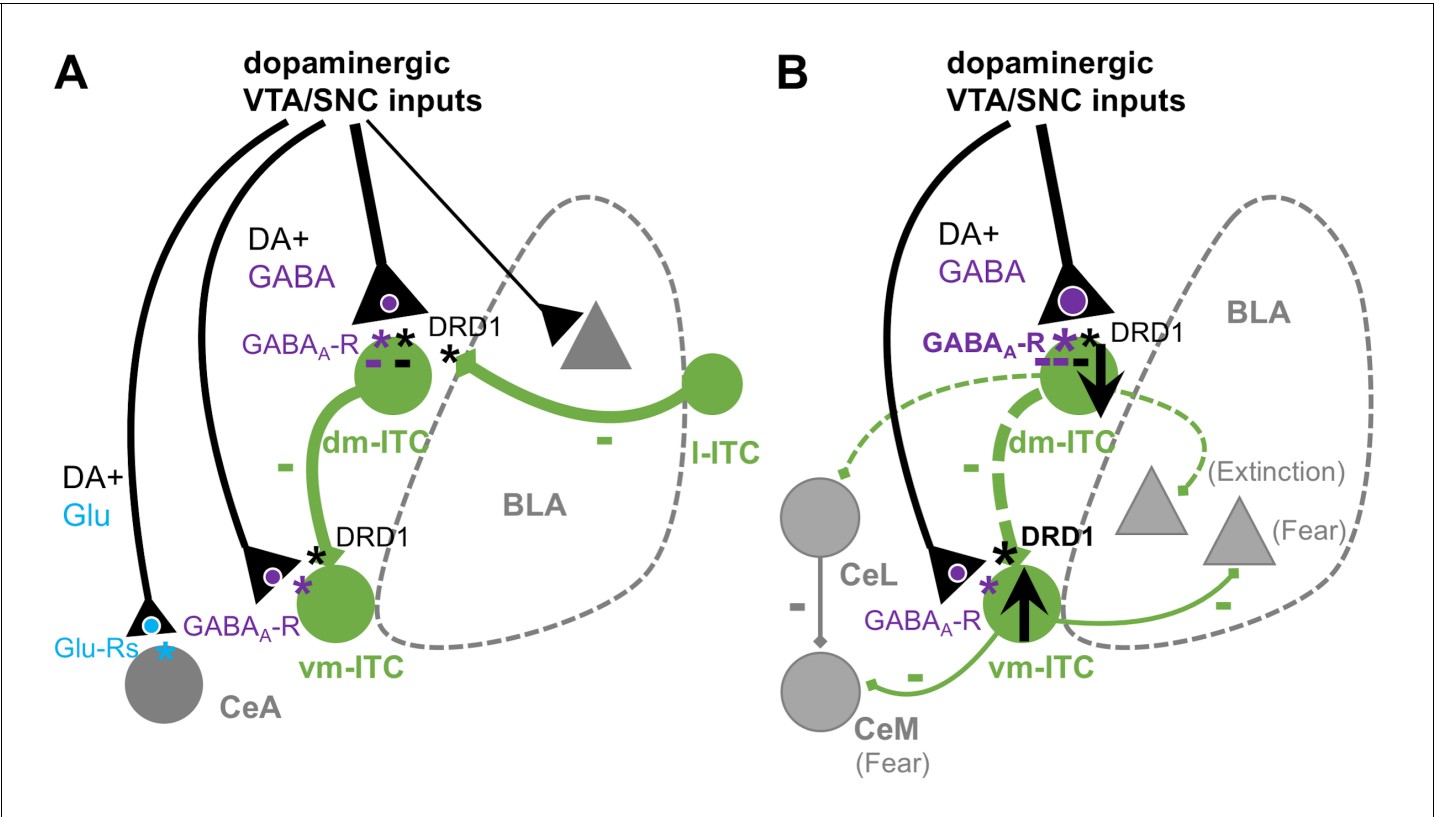

**Figure 7.** Summary scheme of results. (**A**) Impact of dopaminergic midbrain inputs onto intercalated cell (ITC) clusters, central amygdala (CeA), and basal amygdala (BLA). Dopaminergic inputs from ventral tegmental area/substantia nigra pars compacta (VTA/SNC) target amygdala ITC clusters and, to a lesser extent, BLA and CeA. The dorsomedial-intercalated cell (dm-ITC) cluster is more densely and differentially innervated than the ventromedial-intercalated cell (vm-ITC) cluster. Co-release of glutamate is mainly observed in CeA, whereas GABA co-release is prominent onto dm- and vm-ITCs. Released dopamine (DA) directly hyperpolarizes some dm-ITCs, and DA depresses inhibitory interactions between ITC clusters via presynaptic DRD1. Dopaminergic midbrain inputs can thus regulate the ITC network by inhibitory and disinhibitory mechanisms acting on different time scales. (**B**) Effects of early extinction on dopaminergic regulation of ITC cluster activity and interactions. Early extinction training enhances both direct fast inhibition of dm-ITCs, by biasing midbrain inputs toward GABA co-release, and disinhibition of vm-ITCs, by altering DA-mediated suppression of the dm-ITC→vm-ITC pathway. Together, this may tip the activity balance toward decreased dm-ITC and increased vm-ITC activity. We speculate that this could impact behavioral outcome by decreasing inhibition onto dm-ITC targets (such as CeL and extinction-promoting neurons in BLA), while promoting inhibition onto vm-ITC targets (such as fear-promoting neurons in centro-medial amygdala [CeM] and BLA).

medium spiny neurons of the direct and indirect pathways (*Tritsch et al., 2012*). The presence of vGluT1/2 and vGAT in dopaminergic fibers innervating medial ITC clusters is in accordance with canonical release mechanisms for glutamate and GABA, respectively. In dorsal striatum, however, GABA is transported into vesicles via the vesicular monoamine transporter VMAT2, suggesting a spatio-temporal synchronization of GABA and DA release from the same vesicles (*Tritsch et al., 2012*). Which of the two mechanisms is predominant in ITC afferents remains to be investigated. Furthermore, in dopaminergic striatal afferents, GABA is either synthesized in a non-canonical alde-hyde-dehydrogenase1a1-dependent pathway and/or relies on GABA uptake from the extracellular milieu via GABA transporters (*Kim et al., 2015*; *Tritsch et al., 2014*). Although we did not deter-mine the mechanism that provides GABA for inhibiting ITCs, regulation of GABA uptake could endow midbrain afferents with the flexibility to rapidly alter GABAergic co-transmission in a target-specific manner. It may, therefore, be a good candidate mechanism mediating changes in GABA co-release during early extinction learning.

ITCs receive a number of well-characterized excitatory inputs from BLA, thalamus, prefrontal and sensory cortex that drive their activity (*Amir et al., 2011*; *Asede et al., 2015*; *Cho et al., 2013*; *Paré et al., 2003*; *Strobel et al., 2015*). Apart from local inhibitory interactions between ITCs, which

were proposed to stabilize overall spike output (*Geracitano et al., 2007*; *Mańko et al., 2011*; *Royer et al., 2000*), it remained unclear which extrinsic inputs inhibit ITCs. Co-release of GABA from dopaminergic midbrain could provide a rapid and temporally precise signal to inhibit ITC spike activity in response to salient stimuli and/or during learning. Furthermore, spatially targeted inhibition onto ITC dendrites or spines may shunt glutamatergic inputs to enable local control of synaptic plasticity (*Tritsch et al., 2016*).

Phasic activity of midbrain dopaminergic neurons may signal unexpected rewards or aversive events such as noxious stimuli, and these activity patterns shape DA release in target structures (*Brischoux et al., 2009*; *Bromberg-Martin et al., 2010*). Phasic stimulation (>20 Hz) of VTA neurons increases DA release in forebrain structures, including amygdala, and affects behavior (*Holloway et al., 2019*; *Tsai et al., 2009*). In our hands, phasic stimulation of midbrain dopaminergic inputs hyperpolarized dm-ITCs similarly to direct DA application, and depressed sIPSCs in a DA-R-dependent manner, strongly suggesting that physiologically released DA modulates ITC function. Our finding that a fraction of ITCs is directly DA-responsive may help to resolve discrepancies from earlier work demonstrating hyperpolarization of dm- and vm-ITCs in young (*Mańko et al., 2011*; *Marowsky et al., 2005*) versus no detectable effect in dm-ITCs in adult animals (*Kwon et al., 2015*). A reduction of sIPSC amplitude may suggest a postsynaptic mechanism, whereby DA could directly modulate $GABA_A$-Rs to decrease $GABA_A$ currents, similarly to what was observed in striatum (*Flores-Hernandez et al., 2000*; *Hoerbelt et al., 2015*; *Tritsch and Sabatini, 2012*). Given that sIPSCs include action potential-driven events, and considering that release can be multivesicular (*Rudolph et al., 2015*), a reduction in presynaptic input activity or release probability could contribute to a decreased sIPSC amplitude. The novel aspect of our work is that DA also modulates ITC interactions via presynaptic DRD1. Thus, activity of dm-ITCs is shaped by DA modulation of their input from l-ITCs and output to vm-ITCs, as well as direct DA-induced hyperpolarization, indicating that several distinct dopaminergic mechanisms act in concert to gate ITC activity.

While ITCs form local inhibitory networks not only within but also between medially located neighboring clusters (*Amir et al., 2011*; *Busti et al., 2011*; *Geracitano et al., 2007*; *Morozov et al., 2011*; *Royer et al., 2000*), our novel finding that l-ITCs provide input to dm-ITCs suggests further complexity of inhibitory and disinhibitory interactions between ITC clusters that can provide a substrate for integration of information from distinct afferents (*Asede et al., 2015*; *Morozov et al., 2011*; *Strobel et al., 2015*). DA also modulated the interactions between ITC clusters. Its disinhibitory effect via presynaptic DRD1 in the target cluster resembles the situation in the nucleus accumbens, where presynaptic DRD1 attenuates IPSCs and consequently, lateral inhibition between medium spiny neurons, a mechanism proposed to diminish competitive interactions between single projection neurons or ensembles (*Nicola and Malenka, 1997*; *Pennartz et al., 1992*; *Taverna et al., 2005*). In analogy, the same mechanism may enable selection of specific ITC clusters or ensembles within clusters during distinct behavioral states. Modulation of the l-ITC→dm-ITC pathway could thus function to select and/or amplify the output of dm-ITCs with distinct projection patterns (*Asede et al., 2015*; *Busti et al., 2011*; *Duvarci and Pare, 2014*). While postsynaptic inhibition of ITCs would decrease overall inhibitory output onto diverse downstream regions, target-specific presynaptic disinhibition can promote inhibition from distinct ITC clusters or ensembles onto defined downstream regions. These mechanisms may help to sharpen competitive interactions between clusters and to select a defined ITC network output.

In the context of behavior, in vivo pharmacological studies in amygdala pointed to a role of DRD1 in the acquisition of cued fear memory (*Guarraci et al., 1999*; *Lamont and Kokkinidis, 1998*; *Nader and LeDoux, 1999*) as well as in the acquisition of extinction (*Hikind and Maroun, 2008*). Cellular effects of DRD1 include altered excitability of BLA projection and local interneurons, as well as DRD1-dependent long-term potentiation in the BLA-CeA pathway that has been discussed in the context of fear acquisition (*Groessl et al., 2018*; *Lee et al., 2017*). However, DRD1-dependent ITC network modulation could provide the required flexibility to support behavioral transitions during acquisition of extinction memory. Circuit and immediate early gene mapping studies converged on a model where dm-ITCs become engaged and undergo plasticity during fear learning, keeping vm-ITC activity in check. During extinction learning, this balance tips toward recruitment of vm-ITCs, which are required for extinction retrieval (*Busti et al., 2011*; *Duvarci and Pare, 2014*; *Likhtik et al., 2008*). Our data suggest that dopaminergic afferent control over ITCs could play a critical role in these transitions. Consistent with the above model, enhanced GABA co-release onto

dm-ITCs during early extinction could rapidly decrease their activity and, together with enhanced DRD1-mediated disinhibition of vm-ITCs, allow for activation of the latter.

DA neurons in the VTA are activated by unexpected omission of the US, implying that they provide a prediction error-like signal to downstream targets, including ITCs (*Luo et al., 2018*; *Salinas-Hernández et al., 2018*). A recent study indicates that medial versus lateral VTA activity provides a reward-prediction error-like signal and a salience signal, respectively, during extinction learning (*Cai et al., 2020*). Furthermore, emerging evidence starts to point to a role for the nigrostriatal pathway in emotional regulation. Activation of the SNC, for example, enhances extinction memory retrieval and renders it more resistant to renewal (*Bouchet et al., 2018*). Therefore, a differential innervation from dopaminergic VTA/SNC regarding strength and input origin could provide a more prominent modulation of dm- versus vm-ITC activity. This could enable both extinction learning and retrieval by dampening dm-ITC activity more strongly to enable vm-ITC disinhibition.

Downstream dm- and vm-ITCs most likely exert behavioral output by differentially inhibiting specific fear- and extinction-related subpopulations of BLA and CeL, or centro-medial (CeM) neurons (*Figure 7B*; *Asede et al., 2015*; *Duvarci and Pare, 2014*; *Gregoriou et al., 2019*). Concurring or subsequent synaptic plasticity at BLA and other afferents onto dm- and vm-ITCs could help maintain this activation pattern during extinction retrieval (*Amano et al., 2010*; *Asede et al., 2015*; *Huang et al., 2014*; *Kwon et al., 2015*).

While our study delineates several novel cellular mechanisms by which mesolimbic and nigrostriatal dopaminergic afferents control amygdala ITC networks, the observed ex vivo alterations are still correlative. Further studies are warranted to understand how these cellular mechanisms impact fear and extinction learning and memory. Beyond fear memories, DA is critical for processes related to incentive salience, motivation, and cue-reward learning (*Abraham et al., 2014*; *Bromberg-Martin et al., 2010*). For example, VTA dopaminergic inputs to BLA are activated by motivationally salient appetitive and aversive outcomes and acquire responses to predictive cues during learning (*Lutas et al., 2019*). DA is also released in the amygdala during affective states such as stress (*Belujon and Grace, 2015*; *Yokoyama et al., 2005*). Therefore, it is intriguing to speculate that dopaminergic modulation of distinct ITC clusters may also play a role in other processes such as reward learning and addiction, as well as in the control of mood-related behaviors, such as anxiety- and depression-like behavior (*Ferrazzo et al., 2019*; *Kuerbitz et al., 2018*).

## Materials and methods

**Key resources table**

| Reagent type (species) or resource | Designation | Source or reference | Identifiers | Additional information |
|---|---|---|---|---|
| Strain, strain background (*Mus musculus*) | Male C57BL/6J mice | Charles River or Envigo | Stock#: 000664, RRID:IMSR_JAX:000664 | |
| Genetic reagent (*Mus musculus*) | Male heterozygous B6.SJL-*Slc6a3*<sup>tm1.1(cre)Bkmn</sup>/J mice | The Jackson Laboratory | Stock#: 006660, RRID:IMSR_JAX:006660 | Maintained on C57BL/6J background |
| Genetic reagent (*Mus musculus*) | Male heterozygous B6.Cg-*Foxp2*<sup>tm1.1(cre)Rpa</sup>/J mice | The Jackson Laboratory | Stock#: 030541, RRID:IMSR_JAX: 030541 | Maintained on C57BL/6J background |
| Genetic reagent (*Mus musculus*) | Male heterozygous ICR.Cg-*Gad1*<sup>tm1.1Tam</sup>/Rbrc mice | *Tamamaki et al., 2003. J Comp Neurol* 467(1):60–79. doi:10.1002/cne.10905 | RRID:IMSR_RBRC03674 | Maintained on C57BL/6J background |
| Strain, strain background (adeno-associated virus) | AAV2/1.EF1a.DIO.hChR2(H134R).eYFP | Addgene / U. Penn Vector Core | Cat# 20298-AAV1, RRID:Addgene viral prep # 20298-AAV1 | |
| Strain, strain background (adeno-associated virus) | AAV2/9.EF1a.DIO.hChR2(H134R).eYFP | Addgene / U. Penn Vector Core | Cat# 20298-AAV9, RRID:Addgene viral prep # 20298-AAV9 | |

*Continued on next page*

*Continued*

| Reagent type (species) or resource | Designation | Source or reference | Identifiers | Additional information |
|---|---|---|---|---|
| Strain, strain background (adeno-associated virus) | AAV2/1.EF1a. DIO.hChR2 (H134R).mCherry | Addgene/Viral Vector Facility U. Zurich | Cat# 20297-AAV1, RRID:Addgene viral prep # 20297-AAV1 | |
| Antibody | Anti-GFP (goat polyclonal) | GeneTex | Cat# GTX26673, RRID:AB_371426 | IF(1:500) |
| Antibody | Anti-GFP (rabbit polyclonal) | Invitrogen | Cat# A11122, RRID:AB_221569 | IF(1:750 or 1:1000) |
| Antibody | Anti-FoxP2 (rabbit polyclonal) | Abcam | Cat# AB16046, RRID:AB_2107107 | IF(1:1000) |
| Antibody | Anti-FoxP2 (mouse monoclonal) | Merck Millipore | Cat# MABE415, RRID:AB_2721039 | IF(1:1000) |
| Antibody | Anti-FoxP2 (sheep polyclonal) | R and D Systems | Cat# AF5647, RRID:AB_2107133 | IF(1:200), IHC(1:200) |
| Antibody | Anti-TH (rabbit polyclonal) | Millipore | Cat# AB152, RRID:AB_390204 | IF(1:1000), IHC (1:4000) |
| Antibody | Anti-TH (mouse monoclonal) | Merck Millipore | Cat# MAB318, RRID:AB_2313764 | IF(1:1000) |
| Antibody | Anti-Bassoon (mouse monoclonal) | Abcam | Cat# AB82958, RRID:AB_1860018 | IF(1:500) |
| Antibody | Anti-VGAT (guinea pig polyclonal) | Synaptic Systems | Cat# 131 004, RRID:AB_887873 | IF(1:500) |
| Antibody | Anti-VGLUT1 (guinea pig polyclonal) | Chemicon/ Millipore | Cat# AB5905, RRID:AB_2301751 | IF(1:3000) |
| Antibody | Anti-VGLUT2 (guinea pig monoclonal) | Chemicon/ Millipore | Cat# AB5907, RRID:AB_2301731 | IF(1:5000) |
| Antibody | Anti-Goat Alexa 488 (donkey polyclonal) | Invitrogen/Life Technologies | Cat# A-11055, RRID:AB_2534102 | IF(1:1000) |
| Antibody | Anti-Rabbit Alexa 555 (donkey polyclonal) | Invitrogen/Life Technologies | Cat# A-31572, RRID:AB_162543 | IF(1:1000) |
| Antibody | Anti-Mouse Alexa 555 (goat polyclonal) | Invitrogen/Life Technologies | Cat# A-21424, RRID:AB_141780 | IF(1:1000) |
| Antibody | Anti-Rabbit Alexa 647 (donkey polyclonal) | Invitrogen/Life Technologies | Cat# A-31573, RRID:AB_2536183 | IF(1:1000) |
| Antibody | Anti-Mouse Alexa 633 (goat polyclonal) | Invitrogen/Life Technologies | Cat# A-21052, RRID:AB_2535719 | IF(1:1000) |
| Antibody | Anti-Rabbit Alexa 633 (goat polyclonal) | Invitrogen/Life Technologies | Cat# A-21071, RRID:AB_2535732 | IF(1:1000) |
| Antibody | Anti-Rabbit Alexa 488 (donkey polyclonal) | Life Technologies | Cat# A21206, RRID:AB_2535792 | IF(1:1000) |
| Antibody | Anti-Goat Cy5 (donkey polyclonal) | Jackson Immuno-research | Cat# 705-175-147, RRID:AB_2340415 | IF(1:400) |

*Continued on next page*

*Continued*

| Reagent type (species) or resource | Designation | Source or reference | Identifiers | Additional information |
|---|---|---|---|---|
| Antibody | Anti-Guinea pig DyLight 405 (donkey polyclonal) | Jackson Immuno-research | Cat# 706-475-148, RRID:AB_2340470 | IF(1:500) |
| Antibody | Anti-rabbit (goat polyclonal) | Vector Laboratories | Cat# BA-1000, RRID:AB_2313606 | IHC(1:500) |
| Antibody | Anti-goat/sheep (horse polyclonal) | Vector Laboratories | Cat# BA-9500, RRID:AB_2336123 | IHC(1:500) |
| Chemical compound, drug | Tetrodotoxin citrate (TTX) | Alomone Labs | Cat# T-550 | |
| Chemical compound, drug | 4-amino pyridine (4-AP) | Sigma Aldrich | Cat# 275875 | |
| Chemical compound, drug | Picrotoxin (PTX) | Sigma Aldrich | Cat# P1675 | |
| Chemical compound, drug | 6,7-dinitro quinoxaline-2, 3-dione (DNQX) disodium salt | Tocris Bioscience | Cat# 2312 | |
| Chemical compound, drug | DL-2-Amino-5-phosphono pentanoic acid sodium salt (DL-AP5) | BioTrend | Cat# BN0086 | |
| Chemical compound, drug | Dopamine hydrochloride | Sigma Aldrich | Cat# H8502 | |
| Chemical compound, drug | Dihydrexidine hydrochloride (DH) | Tocris Bioscience | Cat# 0884 | |
| Chemical compound, drug | Quinpirole hydrochloride | Sigma Aldrich | Cat# Q102 | |
| Chemical compound, drug | CGP55845 hydrochloride | Tocris Bioscience | Cat# 1248 | |
| Chemical compound, drug | SCH23390 hydrochloride | Tocris Bioscience | Cat# 0925 | |
| Chemical compound, drug | Sulpiride | Tocris Bioscience | Cat# 0895 | |
| Chemical compound, drug | L-745870 trihydro-chloride | Tocris Bioscience | Cat# 1002 | |
| Software, algorithm | SPSS | IBM | RRID:SCR_002865 | |
| Software, algorithm | Igor Pro | WaveMetrics | RRID:SCR_000325 | |
| Software, algorithm | NeuroMatic | NeuroMatic | RRID:SCR_004186 | |
| Software, algorithm | ImageJ | NIH | RRID:SCR_003070 | |
| Software, algorithm | Fiji | Fiji | RRID:SCR_002285 | |

*Continued on next page*

*Continued*

| Reagent type (species) or resource | Designation | Source or reference | Identifiers | Additional information |
|---|---|---|---|---|
| Software, algorithm | pClamp 10 | Molecular Devices | RRID:SCR_011323 | |
| Software, algorithm | ZEN 2009 Software | Zeiss | RRID:SCR_013672 | |
| Software, algorithm | Huygens Suite 19.04 | Scientific Volume Imaging | RRID:SCR_014237 | |
| Software, algorithm | Imaris Software, versions 7.6.1 and 9.7.0 | Oxford Instruments, Bitplane | RRID:SCR_007370 | |
| Other | NeuroTrace stain 435/455 | Thermo Fisher Scientific | Cat# N21479 | |
| Other | Vectastain Elite ABC-Peroxidase kit | Vector Laboratories | Cat# PK-7100, RRID:AB_2336827 | |
| Other | CY3-Streptavidin | Dianova/ Jackson Immuno- Research | Cat# 016-160-084, RRID:AB_2337244 | |
| Other | Pacific Blue- Streptavidin | Thermo Fisher Scientific | Cat# S11222 | |

## Animals

We used 6- to 14-week-old adult male B6.SJL-*Slc6a3*$^{tm1.1(cre)Bkmn}$/J mice (JAX#006660, Jackson Laboratories, Bar Harbor, Maine; *Bäckman et al., 2006*), referred to as DAT-Cre mice, or B6.Cg-*Foxp2*$^{tm1.1(cre)Rpa}$/J mice (JAX#030541, Jackson Laboratories; *Rousso et al., 2016*), referred to as FoxP2-Cre mice, for Cre-dependent expression of viral vectors. Young (20- to 28-day-old) or adult (8- to 10-week-old) male GAD67–GFP (ICR.Cg-Gad1tm1.1Tam/Rbrc mice, *Tamamaki et al., 2003*) mice were used for slice experiments without optical stimulation. Adult (12-week-old) wild-type C57BL/6J male mice (Charles River, Sulzfeld, Germany) were used for quantification of TH staining. All transgenic lines were heterozygous and backcrossed to C57BL/6J. Mice were kept in a 12-hr light/dark (6 am to 6 pm) cycle with access to food and water ad libitum. All behavioral experiments were conducted during the light cycle (between 7 and 10 am). All animal procedures were performed in accordance with institutional guidelines and with current European Union guidelines, and were approved by the local government authorities for Animal Care and Use (Regierungspraesidium Tuebingen, State of Baden-Wuerttemberg, Germany and the Austrian Animal Experimentation Ethics Board BMWFW-66.011/0021-WF/V/3b/2016).

## Surgical procedures

Stereotaxic injection of AAV-EF1a-DIO-hChR2(H134R)-eYFP (serotype 2/1 or 2/9, U. Penn Vector Core, Philadelphia, PA, or Addgene, Watertown, MA) was performed in 6- to 8-week-old DAT-Cre or FoxP2-Cre transgenic mice. Mice were anesthetized with isoflurane in oxygen-enriched air (Oxymat 3, Weinmann Medical Technologies, Hamburg, Germany), and the head was fixed in a stereotaxic frame (Kopf Instruments, Tujunga, CA or Stoelting, Wood Dale, IL). Eyes were protected with ointment, and the body temperature of the animal was maintained using a feedback-controlled heating pad with a rectal sensor (FHC, Bowdoin, ME). Lidocaine was used as a local anesthetic. An incision was made on the skin, the skull was exposed, and coordinates of bregma were identified. The skull was drilled with a microdrill (Kopf Instruments, Tujunga, CA) at the desired coordinates in reference to bregma. Borosilicate capillaries for VTA/SNC (1B150F-4; World Precision Instruments, Friedberg, Germany) and ITC injections (marked 1–5 µl; Drummond Scientific, Broomall, PA) were pulled on a horizontal pipette puller (P-1000; Sutter Instruments, Novato, CA) and used to pressure inject viruses at a volume of 300–500 nl for VTA/SNC and 25–50 nl for ITCs. The capillaries were slowly withdrawn, the skull disinfected, and the skin sutured with silk. Postoperative pain medication

included injection of meloxicam (Metacam; Boehringer Ingelheim, Ingelheim, Germany) at 5 mg/kg subcutaneously. The following coordinates were used in reference to bregma (in mm): for the VTA/SNC, AP: −3.00, ML: ±0.50, DV: 4.50; for the dm-ITC cluster, AP: −1.40, ML: ±3.3, DV: 4.70; for the l-ITC cluster, AP: −1.40, ML: ±3.45, DV: 4.70. VTA/SNC dual viral injections were performed with AAV-EF1a-DIO-hChR2(H134R)-eYFP (serotype 2/1; U. Penn Vector Core, Philadelphia, PA or Addgene, Watertown, MA) and AAV-EF1a-DIO-hChR2(H134R)-mCherry (serotype 2/1; Viral Vector Facility, University of Zurich, Switzerland) at the following coordinates in reference to bregma (in mm): for VTA, AP: −3.00, ML: ±0.20, DV: 4.50; for SNC, AP: −3.00, ML: ±1.75, DV: 4.50.

## Slice recordings and analysis

Three (for ITCs) to six (for VTA/SNC) weeks after viral injections, mice were deeply anesthetized with 3% isoflurane (Isofluran CP; cp-pharma, Burgdorf, Germany) in oxygen and decapitated. The brain was rapidly extracted and cooled down in ice-cold slicing artificial cerebrospinal fluid (ACSF) containing (in mM) 124 NaCl, 2.7 KCl, 26 NaHCO$_3$, 1.25 NaH$_2$PO$_4$, 10 MgSO$_4$, 2 CaCl$_2$, 18 D-Glucose, and 4 ascorbic acid, equilibrated with carbogen (95% O$_2$/5% CO$_2$). Coronal brain slices (320 µm) containing the amygdala were cut in ice-cold slicing ACSF with a sapphire blade (Delaware Diamond Knives, Wilmington, DE) on a vibrating microtome (Microm HM650V; ThermoFisher Scientific, Dreieich, Germany). Slices were collected in a custom-built interface chamber with recording ACSF containing (in mM) 124 NaCl, 2.7 KCl, 26 NaHCO$_3$, 1.25 NaH$_2$PO$_4$, 1.3 MgSO$_4$, 2 CaCl$_2$, 18 D-Glucose, and 4 ascorbic acid, equilibrated with carbogen. Slices were recovered at 37°C for 40 min and stored at room temperature. Whole-cell patch-clamp recordings were performed in a submersion chamber under an upright microscope (Olympus BX51WI; Olympus Germany, Hamburg, Germany), where slices were superfused with recording ACSF at 30–31°C. Recordings were performed using an Axon Instruments Multiclamp 700B amplifier and a Digidata 1440A digitizer (both, Molecular Devices, San Jose, CA). Glass micropipettes (6–9 MΩ resistance when filled with internal solution) were pulled from borosilicate capillaries (ID 0.86 mm, OD 1.5 mm; Science Products, Hofheim, Germany).

Recordings in voltage clamp configuration were performed with cesium-based internal solution containing (in mM) 115 Cs-methanesulphonate, 20 CsCl, 4 Mg-ATP, 0.4 Na-GTP, 10 Na$_2$-phosphocreatine, 10 HEPES, and 0.6 EGTA (290–295 mOsm, pH 7.2–7.3). Signals were low-pass filtered at 2 kHz and digitized at 5 kHz. Recordings in current clamp configuration were performed with K-gluconate-based internal solution containing (in mM) 130 K-gluconate, 5 KCl, 4 Mg-ATP, 0.4 Na-GTP, 10 Na2-phosphocreatine, 10 HEPES, and 0.6 EGTA (290–295 mOsm, pH 7.2–7.3). Signals were low-pass filtered at 10 kHz and digitized at 20 kHz. Changes in series resistance <30% were accepted. Optical stimulation was achieved by triggering a light-emitting diode (LED; 470 nm, KSL70; Rapp Opto-Electronics, Hamburg, Germany or CoolLED pE, CoolLED, Andover, UK) coupled to the upright microscope objective (Olympus Germany; 60x/1.0 NA). PSCs were recorded from LED stimulations (pulse length 0.2–5 ms) with 0.1 Hz frequency. Spontaneous and evoked IPSCs were recorded at 0 mV holding potential. For sEPSC recordings, GABA$_A$ receptors were blocked with PTX (100 µM) and cells were held at −70 mV. Phasic stimulation of dopaminergic fibers consisted of 10 pulses at 30 Hz repeated 10 times with 10 s inter-sweep interval. Tonic stimulation consisted of a single sweep of 100 pulses at 1 Hz. Hyperpolarization was assessed in current clamp mode, either via bath application of DA in the presence of PTX (100 µM), or upon optogenetic stimulation with 10 pulses at 30 Hz in the presence of GABA and glutamate receptor blockers as indicated. In some experiments, biocytin (3–5%) was added to the internal solution for post-hoc visualization of the recorded cells.

Electrophysiological data were analyzed with Neuromatic (http://www.neuromatic.thinkrandom.com/; *Rothman and Silver, 2018*) and/or custom written functions in Igor Pro (Wavemetrics, Portland, OR). PSC amplitudes were measured as the peak value in reference to a 5 ms baseline period. For co-release data, reversal potential (Erev) was calculated by regression analysis of the peak current at different holding potentials (−70, −50, 0, and +40 mV). Cells with Erev < -40 mV were considered to have GABAergic, −40 mV < Erev < -15 mV to have mixed, and Erev > -15mV to have glutamatergic PSCs. For DA-induced hyperpolarization data in young and adult animals, the membrane potential was measured as the maximum hyperpolarization averaged within a 2- to 3-min time window relative to a 2- to 3-min baseline before DA application. For dopaminergic fiber stimulation, the maximum membrane potential change (averaged over 100 ms) was detected relative to 500 ms baseline and recovery periods, within a 2 s window post stimulation.

Chemicals were purchased from Carl Roth (Karlsruhe, Germany) or Merck/Sigma-Aldrich (Darmstadt, Germany). Drugs for pharmacology were obtained from Sigma-Aldrich (Quinpirole, DA, PTX, 4-AP), Tocris Bioscience/BioTeche (Wiesbaden, Germany) (DH, CGP55845, SCH23390, Sulpiride, L-745870, DNQX), or Biotrend (Cologne, Germany) (DL-AP5). All drugs were diluted in ACSF from concentrated frozen stocks on the day of recording. Drugs were bath-applied at a rate of 2 ml/minute using a peristaltic pump (Ismatec Products, Cole-Palmer, Wertheim, Germany).

## Behavioral procedures

Fear conditioning was performed in a square chamber with a grid floor that was cleaned with 70% ethanol (context A). Five pairings of conditioned (CS) and unconditioned (US) stimuli were presented. The CS was a 30 s tone (at 7.5 kHz, 80 dB), which coincided at its offset with the US (1 s foot shock, 0.4 mA). In the CS-only group, the US was omitted. Fear extinction was performed in a round chamber with a flat floor that was cleaned with 1% acetic acid (context B). Early extinction training started 24 hr after fear conditioning and consisted of 16 CSs. Full extinction training was performed by presenting 25 CSs 24 and 48 hr after conditioning. All sessions started with a 2-min baseline assessment, and the subsequent CSs were presented at random intervals of 20–180 s. Freezing was detected using an infrared beam detection system (Coulbourn Instruments, Holliston, MA), with the threshold for freezing set to 2 s of immobility and quantified offline with custom-written macros in Microsoft Excel (*Asede et al., 2015*). Freezing data for fear conditioning are shown as average post-shock freezing for the last two CS-US pairings. CS-induced freezing during extinction is shown as averages of four CSs. Animals were randomly assigned to experimental groups. Electrophysiology experiments were performed 80–90 min after the behavioral procedures by an experimenter blinded to the training procedure of the animal.

## Immunohistochemistry, imaging procedures, and image analysis

Slices containing ITC clusters and midbrain injection sites were fixed in 4% paraformaldehyde (PFA) in phosphate buffered saline (PBS), whereas recorded slices with biocytin-filled cells were fixed in 4% PFA, 15% saturated picric acid solution, and 0.05% glutaraldehyde in PBS overnight at 4°C. Recorded slices were resectioned on a vibratome (Microm HM650V; ThermoFisher Scientific, Dreieich, Germany) at 60–65 µm and processed as previously described (*Asede et al., 2015*) with minor modifications.

Biocytin was revealed using fluorescently conjugated Streptavidin (Dianova, Germany or ThermoFisher Scientific; 1:1000). Immunostainings were performed using standard procedures. Upon blocking with PBS complemented with 0.3% Triton and 10% serum for 90 min, the following primary antibodies were used: goat anti-GFP (GeneTex, 1:500), rabbit anti-GFP (Invitrogen, 1:750), rabbit anti-FoxP2 (Abcam, 1:1000), mouse anti-FoxP2 (Merck Millipore, 1:1000), rabbit anti-TH (Millipore, 1:1000), mouse anti-TH (Merck, 1:1000), and mouse anti-Bassoon (Abcam, 1:500). Secondary antibodies used were Alexa488-conjugated donkey anti-rabbit, Alexa555-conjugated donkey anti-rabbit and goat anti-mouse, Alexa647-conjugated donkey anti-rabbit, and Alexa633-conjugated goat anti-mouse and goat anti-rabbit. All secondary antibodies were obtained from Invitrogen/Life Technologies and used in 1:1000 dilutions. Some sections were counterstained with NeuroTrace 435/455 (ThermoFisher Scientific, 1:500).

Overview images were taken with a fluorescence microscope (Axio Imager; Carl Zeiss, Oberkochen, Germany). Detailed images of injection and projection sites were taken with a LSM710 laser-scanning microscope (Carl Zeiss, Oberkochen, Germany) equipped with either a 40x/1.3 NA or a 63x/1.4 NA objective, with the pinhole set to 1 airy unit. Most of the confocal images are presented as maximum intensity projection of z-stacks. For deconvolution, Huygens software (Scientific Volume Imaging, Hilversum, The Netherlands) was used. Cell counts were performed from confocal z-stack images with the cell counter plugin FIJI in ImageJ (https://imagej.net/Cell_Counter). Specificity of the infections in the dopaminergic midbrain was calculated as the ratio of double-labeled cells (TH+GFP+ cells) to all infected cells (GFP+ cells).

For analysis of presynaptic markers, 60 µm coronal sections, containing the ITCs from DAT-Cre mice injected with AAV-EF1a-DIO-hChR2(H134R)-eYFP, were immunostained for vGluT1/2 or vGAT. Sections were incubated with the following primary antibodies: polyclonal rabbit anti-GFP (Invitrogen, 1: 1000), sheep anti-FoxP2 (R and D Systems, 1:200), and polyclonal guinea pig anti-vGAT

(Synaptic Systems, 1:500) or a combination of guinea pig anti-vGluT1 (Millipore, 1:3000) and guinea pig anti-vGluT2 (Chemicon, 1:5000). Secondary antibodies used were Alexa488-conjugated donkey anti-rabbit (Life Technologies, 1:1000), Cy5-conjugated donkey anti-goat (Jackson Immunoresearch Lab, 1:400), and DyLight405-conjugated donkey anti-guinea pig (Jackson Immunoresearch Lab, 1:500). Images of vGluT and vGAT stainings were visualized using an Airy Scan LSM980 laser scanning microscope (Carl Zeiss, Oberkochen, Germany) with a 40x/1.3 NA objective. The pinhole was set to one airy unit. Z-stacks were obtained and analyzed in 3D using Imaris 7.6.1 Software (Bitplane, Zurich, Switzerland).

3D imaging and quantification of ChR2-YFP+ axons within dm- and vm-ITC clusters was performed on 1 image volume per animal. The image volume was selected for the highest apparent density of axons among several scans (n = 3–4 per animal) and for the presence of FoxP2-labeled neurons throughout the whole volume. The sections were taken between bregma levels −1.40 and −1.80. Images were acquired using an Airy Scan LSM980 laser scanning microscope (Carl Zeiss, Oberkochen, Germany) with a 40x/1.2 NA objective (using x3 zoom) or a Leica TCS SP8 gSTED microscope equipped with an HCX PL APO 63x/1.3 NA objective (Leica Microsystems GmbH, Germany). Raw images were channel dye separated and deconvolved using Huygens software (Scientific Volume Imaging, Hilversum, The Netherlands). For 3D measurements, IMARIS 9.7.0 software (Oxford Instruments, Bitplane, Zurich, Switzerland) was used. The Surpass function was applied to generate surfaces of the ChR2-YFP+ fibers (*Figure 1—figure supplement 3*). The volume covered by the surfaces was calculated in $\mu m^3$ and expressed as % of the total volume.

For the quantification of TH-IR, coronal sections were cut (40 µm) on a Leica VT1000S vibratome (Leica Microsystems, Vienna, Austria) and serial sections containing ITC clusters immunostained against TH or FoxP2 according to previously published procedures (*Sreepathi and Ferraguti, 2012*). A rabbit anti-TH (Millipore, 1:4000) and a sheep anti-FoxP2 (R and D Systems, 1:200) antibody were used and diluted in 2% normal serum (goat for anti-TH and horse for anti-FoxP2), 0.3% Triton X-100 in Tris-buffered saline (TBS; pH 7.4). Biotinylated secondary antibodies (goat anti-rabbit and horse anti-sheep; Vector Laboratories; both 1:500) were applied overnight, and the antigen-antibody complex was visualized by the avidinbiotin-horseradish peroxidase procedure (Vectastain Elite ABC kit; Vector Laboratories) using 3,3'-diaminobenzidine. Digital images were taken at x5 magnification, which allowed to have both the dm- and vm-ITC clusters in the same image, using an Axiophot microscope (Carl Zeiss, Oberkochen, Germany) equipped with an AxioCam MRc5 camera. Quantitative evaluation of the relative optical density (ROD) was performed with ImageJ 1.53a software (NIH, USA). The outlines of the dm- and vm-ITC clusters were based on the FoxP2 immunoreactivity observed in consecutive sections. The dm- and vm-ITC cluster ROD was determined as the ratio between their mean grey value (MGV) and the MGV of a reference area (central subdivision of the central nucleus of the amygdala), subtracted of the background MGV (measured within the optic nerve tract) (*Figure 1—figure supplement 4*).

## Data presentation and statistics

Most data are represented as individual points, box and whisker plots (reporting median, 25, 75, 10, and 90 percentiles), or bar graphs of average ± SEM. Where applicable, electrophysiology data were normalized to the baseline period. Z-scores were calculated from the average and standard deviation of the baseline.

No methods were used to predetermine sample sizes. Sample sizes are similar to those in other studies in the field. Statistical analysis was performed using the program SPSS (IBM, Germany). A p-value<0.05 was considered significant. Paired or unpaired Student's t-tests were used for two dependent or independent comparisons of continuous data, respectively. Wilcoxon or Mann-Whitney tests were used for paired or unpaired comparisons of scaled data, respectively. Comparison of multiple groups was done using one- or two-way ANOVA followed by Bonferroni or Tukey's multiple comparison post-hoc tests, as appropriate. Repeated treatments were compared using repeated-measures ANOVA followed by a Bonferroni post-hoc test, or Bonferroni-corrected pairwise comparisons as indicated. Categorical data were compared with Fisher's exact test.

## Acknowledgements

We thank Johannes Ungermann and Marlly Achury for help with histology, members of the Ehrlich and Ferraguti labs for discussion and support, and Dr. Julien Genty, Martin Zeller, and Prof. Norbert Hajos for critical reading and commenting on an earlier version of the manuscript. We also thank Dr. Martin Offterdinger from the Bioptics core facility of the Medical University of Innsbruck for helping with volume measurements. This work was supported by the Charitable Hertie Foundation and the German Research Foundation DFG EH 197/3–1 (to IE) and by the Austrian Science Fund (FWF) grant # I-2215 (to FF).

## Additional information

### Funding

| Funder | Grant reference number | Author |
|---|---|---|
| Hertie Foundation | | Ingrid Ehrlich |
| Deutsche Forschungsge-meinschaft | DFG EH 197/3-1 | Ingrid Ehrlich |
| Austrian Science Fund | I-2215 | Francesco Ferraguti |

The funders had no role in study design, data collection and interpretation, or the decision to submit the work for publication.

### Author contributions

Ayla Aksoy-Aksel, Conceptualization, Data curation, Formal analysis, Investigation, Visualization, Methodology, Writing - original draft, Writing - review and editing; Andrea Gall, Formal analysis, Investigation, Visualization, Methodology, Writing - review and editing; Anna Seewald, Investigation, Visualization, Methodology, Writing - review and editing; Francesco Ferraguti, Conceptualization, Resources, Supervision, Funding acquisition, Investigation, Visualization, Methodology, Writing - review and editing; Ingrid Ehrlich, Conceptualization, Resources, Data curation, Formal analysis, Supervision, Funding acquisition, Investigation, Methodology, Writing - original draft, Project administration, Writing - review and editing

### Author ORCIDs

Ayla Aksoy-Aksel https://orcid.org/0000-0003-0418-6466
Francesco Ferraguti http://orcid.org/0000-0002-3843-5857
Ingrid Ehrlich https://orcid.org/0000-0001-9078-2429

### Ethics

Animal experimentation: All animal procedures were performed in accordance with institutional guidelines and with current European Union guidelines, and were approved by the local government authorities for Animal Care and Use (Regierungspräsidium Tübingen, State of Baden-Württemberg, Germany, Permit Number CIN 2/17 and the Austrian Animal Experimentation Ethics Board - BMWFW-66.011/0021-WF/V/3b/2016). Every effort was made to minimize suffering of the animals.

### Decision letter and Author response

Decision letter https://doi.org/10.7554/eLife.63708.sa1
Author response https://doi.org/10.7554/eLife.63708.sa2

## Additional files

### Supplementary files

- Transparent reporting form

## Data availability

Data deposited in Dryad Digital Repository (https://doi.org/10.5061/dryad.jh9w0vt9b).

The following dataset was generated:

| Author(s) | Year | Dataset title | Dataset URL | Database and Identifier |
|---|---|---|---|---|
| Aksoy-Aksel A, Gall A, Seewald A, Ferraguti F, Ehrlich I | 2021 | Midbrain dopaminergic inputs gate amygdala intercalated cell clusters by distinct and cooperative mechanisms in male mice | https://doi.org/10.5061/dryad.jh9w0vt9b | Dryad Digital Repository, 10.5061/dryad.jh9w0vt9b |

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
