## [Decision Letter]

**Acceptance summary:**

This paper dissects neural circuitry connecting midbrain dopaminergic neurons to inhibitory cell clusters in the amygdala of male mice. The work suggests a novel disinhibitory mechanism for behavioral suppression during conditioned fear extinction, and will be of interest to many in the learning and memory field.

**Decision letter after peer review:**

[Editors’ note: the authors submitted for reconsideration following the decision after peer review. What follows is the decision letter after the first round of review.]

Thank you for submitting your work entitled "Midbrain dopaminergic inputs gate amygdala intercalated cell clusters by distinct and cooperative mechanisms" for consideration by *eLife*. Your article has been reviewed by 3 peer reviewers, and the evaluation has been overseen by a Reviewing Editor and a Senior Editor. The following individual involved in review of your submission has agreed to reveal their identity: Ekaterina Likhtik (Reviewer #3).

Our decision has been reached after consultation between the reviewers. Based on these discussions and the individual reviews below, we regret to inform you that your work will not be considered further for publication in *eLife*.

The reviewers found the work to be of interest, but conveyed that the heavy reliance on supraphysiological manipulations and selective focus on presynaptic mechanisms made the model less convincing. They recommend substantial additional experiments to flesh out these issues, but as a rule, *eLife* does not consider resubmitted manuscripts that require extensive revisions such as these. You are of course welcome to conduct these experiments and submit your work to *eLife* as a new manuscript, however. We hope that you will find the reviewers' comments helpful as you prepare to move forward with your work.

*Reviewer #1:*

This manuscript explores the midbrain dopaminergic influence onto amygdala microcircuitry in male mice. Using optogenetics, electrophysiology, and imaging, the authors demonstrate functional synaptic contacts between midbrain DAergic neurons and amygdalar neurons, particularly in the intercalated cell (ITC) clusters. Evidence of GABA or glutamate co-release from these terminals, as well as DA modulation of inhibitory neurotransmission and alterations in electrophysiological signature following fear conditioning and extinction, are documented. Overall, the manuscript benefits from extensive studies and contributes novel information to the field. There are some issues and questions to be addressed, as outlined below:

1. Are there anatomical (e.g., along the anterior-posterior axis) or physiological factors that correlate with any of the reported observations? This applies to the several experiments reported, e.g.:

a. vGluT_1/2_ or vGAT coexpression on TH^+^ terminals

b. Response of dmITC, vmITC, or CeA neurons to optogenetic stimulation of midbrain DA neurons

c. The fraction of dm-ITCs directly hyperpolarized by DA

2. The authors report that virus injections were localized to the VTA, with less expression in the neighboring SNc. While it is difficult experimentally to isolate these two regions during viral injections, the authors can speculate in the discussion whether the DA input to amygdala subregions from the VTA and SN may play a distinct role in behavior.

a. Additionally, what is known about the localization of amygdalar-projecting midbrain neurons? Are these predominately in the VTA or SN? Is the expression pattern (e.g., location along A-P or M-L axis) similar for TH/GABA or TH/Glu neurons?

3. In what percentage of neurons from Figure 2 tested with glutamate and GABA blockers?

*Reviewer #2:*

This manuscript describes the results of experiments exploring the effects of dopaminergic modulation of neurotransmission between GABAergic ITC clusters in the amygdala, attempting to link the observed neuromodulatory mechanisms to fear behavior. In my view, however, there are certain problems with experimental design in this study. My enthusiasm for this work is somewhat diminished for the reasons outlined below.

Most of the functional studies described in the manuscript were performed with exogenously-applied DA. The only results in the paper potentially showing the physiological effects of endogenously-released DA were obtained in experiments assaying an effect of optogenetic stimulation of dopaminergic fibers on sIPSCs in slices. However, the stimulation episode was fairly short (10 pulses at 30 Hz) and I was not able to find any evidence in the manuscript that this stimulation, in fact, resulted in release of DA. DA-induced hyperpolarization was observed in a fraction of recorded dm-ITC cells (Figure 5, figure supplement 1B-D) with exogenously-applied DA. The authors could perform analogous measurements, looking at effects of phasic stimulation of DA fibers in the presence of glutamatergic and GABAergic antagonists on membrane polarization in ITCs. If the change in membrane polarization (e.g., hyperpolarization) is observed and it is blocked by the D1 receptor antagonist, it would provide direct evidence that photostimulation did induce DA release in studied projections in quantities sufficient to produce measurable changes in ITC functions.

If the amplitude of spontaneous IPCSs is decreased in both dm-ITCs and vm-ITCs following phasic optogenetic activation of dopaminergic fibers, it might be helpful if the authors could speculate how these apparently postsynaptic decreases in GABAergic synaptic currents could be mediated. Moreover, the authors demonstrated nicely that dopaminergic projections to ITCs also release GABA and a fraction of them co-release glutamate. Is there any evidence that the observed effects were mediated by DA but not other neurotransmitters (glutamate or GABA, or both)?

The results of experiments showing that DA may modulate inhibitory neurotransmission between ITC clusters (Figure 4) are inconclusive. The reported suppression of IPSCs was observed when DA was applied exogenously in a fairly high concentration (30 uM). As this concentration might be non-physiologically high, it might be necessary to show that DA released endogenously (e.g., in response to photostimulation of dopaminergic fibers with the same stimulation protocol that was used to modulate sIPSCs) can, in fact, suppress GABAergic synaptic transmission between ITC clusters. Again, the authors were able to activate dopaminergic inputs to ITC clusters optogenetically. Thus, it remains unclear why the mentioned experiments with 'endogenous' dopamine were not performed

The experiments demonstrating that DA-induced suppression of GABAergic synaptic transmission between ITC clusters may be mediated by D1 receptors are inconclusive for the same reason. The observed inhibitory effect of a specific D1 receptor agonist on evoked IPSCs does not necessarily mean that it could be observed under more physiological conditions. Ideally, these experiments should be performed with endogenously released DA to show that photostimulation-induced suppression of IPSCs between clusters is sensitive to D1 receptor antagonists.

The results of extinction experiments are somewhat confusing. The 'early extinction' protocol did not result in any extinction at all. Therefore, the meaning of observed changes in inhibitory synaptic responses in between-clusters connections after extinction training and their sensitivity to dopaminergic modulation is unclear. The observed dissociation between behavioral and network-level effects does not support the notion that the observed neuromodulatory actions of DA on inter-ITC functional connectivity are functionally relevant.

*Reviewer #3:*

This manuscript investigates how the intercalated cell (ITC) islands of the amygdala are modulated by dopamine, and during extinction. This work is motivated by previous findings demonstrating dopamine release from the VTA during early extinction -when prediction error is high – and dopamine receptor expression on the ITC. The authors first demonstrate that VTA/SNc dopaminergic cells projects to the ITC and co-release GABA, resulting in inhibitory postsynaptic potentials. Using in-vitro recordings, the authors then show that phasic stimulation of dopaminergic input decreases the amplitude of spontaneous inhibitory post-synaptic currents (IPSC) in the dorsomedial and ventromedial ITC. Bath application of dopamine also decreased the amplitudes of evoked IPSCs in the ITC, and increase the Paired Pulse Ratio (PPR) in ITC island-to-island signaling in a D1 receptor dependent manner, suggesting that dopamine mediates presynaptic depression in intra-ITC communication via presynaptic D1 receptor activation. The authors then show that after animals undergo early extinction training, stimulation of dopaminergic inputs from the VTA/SNc weakens the evoked IPSCs in the ITC, and enhances presynaptic depression in ITC-to-ITC signaling relative to tone-only controls. The authors suggest that dopaminergic modulation of intra-ITC signaling, from the dorsomedial to the ventromedial ITC island, mediates disinhibition of the ventromedial ITC, leading to their subsequent inhibition of amygdala output. This work makes an important contribution to the field by providing an interesting mechanism for dopamine-mediated modulation of intra-ITC signaling as a function of extinction learning.

1. The authors argue that dopaminergic inputs to the ITC have several effects; they weaken spontaneous (and evoked) ITC activity via GABAergic co-release during phasic dopaminergic activity, and weaken dmITC -to-vmITC (and lITC-to-dmITC) signaling by presynaptic depression of this connection. Both of these factors are proposed to contribute to decreased lateral-to-medial inhibition of ITC islands, in particular during periods of early extinction training. Although, this is an elegant model that is supported by the data, the data also suggest that other dopaminergic influences are at play, but are disregarded by the authors' model as inconsequential. For example, the authors find a direct, postsynaptic influence of dopamine on dmITC, whereby 50% of the neurons are directly hyperpolarized by DA. This is described in the text as only a fraction of neurons exhibiting this response (pg. 12, line 256), and the conclusion is made that DA is acting "chiefly by presynaptic DRD1 receptors" (pg. 12, lines258-9), and again referred to as only a fraction in the Discussion section (pg. 16, line 368). This stress on the presynaptic DRD1 mechanism is also repeated in the model shown in figure 7. Given that half the cells are inhibited by DA post-synaptically, this mechanism should also be incorporated into the model of how DA regulates the ITC.

2. Figure 1 shows DA inputs to both dmITC and vmITC. Given that half of dmITC are directly inhibited by DA, it would be useful to know whether the vmITC are also post-synaptically inhibited by DA. The effect of DA on vmITC pertains strongly to the extinction model proposed by the authors, because it predicts that vmITC are released from inhibition via DA through presynaptic action, whereas there it's quite possible that DA directly hyperpolarizes vmITC. It's not clear if these cells would also be inhibited with phasic stimulation (only dmITC results are shown in supplementary figure 5). It's possible that there are fewer inputs from VTA/SNc onto the vmITC than the dmITC, such that even if DA directly inhibits the vmITCs, there aren't as many inputs to them. In the staining shown in figure 1 Supp2, it appears that the VTA/SNc inputs to the dmITC are much stronger than the they are to the vmITC, however this is not quantified, so it's hard to tell if this is just the example. Can the authors quantify DA input fluorescence to the dmITC/ vmITC in order to get an approximation of the strength of DA input to shed light on this question. If DA directly inhibits these cells, then it's likely that they may become more inhibited after extinction training rather than disinhibited, as in the proposed model.

3. Similarly to the point above, another way that this issue could be addressed is to show that vmITC are more excitable post-extinction compared to the controls.

4. It's not clear what role, if any, the authors believe that the lITC-dmITC connection plays in the model. On the one hand the VTA/SNc connection to the lITC is not assessed neuroanatomically (e.g. it's not shown in figure 1, nor is it drawn in the model in figure 7), or physiologically (e.g. there aren't recordings showing the effects of phasic DA stimulation on lITC islands). On the other hand, the lITC-dmITC connection is shown to undergo PPR depression with DA, similarly to the dmITC-vmITC connection. However, then the question of whether this connection changes with early extinction is not investigated, and not discussed.

5. There are a number of instances where the representative examples are confusing because they don't look like the effect that the authors are hoping to demonstrate. For example, in Figure 3 Supp2, the data show that the frequency of sIPSC at baseline and after stimulation doesn't change, whereas the representative example shows that the frequency of the sIPSCs decreases after stimulation. Similarly, in figure 4I, the example traces on the right are meant to show that DA application leads to increased PPR in lITC -to-dmITC synapse, however the PPR traces don't appear to show that phenomenon. The same is true of the lITC-dmITC PPR in figure 5d (DH drug condition doesn't look like the PPR increases, whereas the group data show that it does). The examples and data shown in figures 6b-c are confusing in terms of color (the reference in the legend is a grey color that isn't on the figure), but also in terms of the finding where the extinction and tone controls are mixed up in order.

6. In some instances, the authors don't report the number of mice from which cells were obtained (e.g. n=6 cells and n=3 cells on pg. 10, lines 211 and 212). Are these cell numbers obtained from multiple mice? It would be helpful to see this data shown as group data, as well as the example data shown.

---

## [Author Response]

[Editors’ note: The authors appealed the original decision. What follows is the authors’ response to the first round of review.]

Reviewer #1:This manuscript explores the midbrain dopaminergic influence onto amygdala microcircuitry in male mice. Using optogenetics, electrophysiology, and imaging, the authors demonstrate functional synaptic contacts between midbrain DAergic neurons and amygdalar neurons, particularly in the intercalated cell (ITC) clusters. Evidence of GABA or glutamate co-release from these terminals, as well as DA modulation of inhibitory neurotransmission and alterations in electrophysiological signature following fear conditioning and extinction, are documented. Overall, the manuscript benefits from extensive studies and contributes novel information to the field.

We thank the reviewer for the positive assessment of our work. We hope that our additional experiments, analyses and text editions can fully address his/her questions.

There are some issues and questions to be addressed, as outlined below:1. Are there anatomical (e.g., along the anterior-posterior axis) or physiological factors that correlate with any of the reported observations? This applies to the several experiments reported, e.g.:a. vGluT_1/2_ or vGAT co-expression on TH^+^ terminalsb. Response of dmITC, vmITC, or CeA neurons to optogenetic stimulation of midbrain DA neuronsc. The fraction of dm-ITCs directly hyperpolarized by DA

We now added two new figure supplements to address these questions:

a. Presynaptic markers were analyzed in dopaminergic terminals to provide proof of principle of their expression. We believe that a comprehensive anatomical analysis will always fall short in the light of the more powerful functional physiological analysis of corelease of glutamate and/or GABA (see b).

b. We have retrieved the locations for all recorded neurons in dm-ITC, vm-ITC, and CeA in slices from animals that showed co-release. We find that cells receiving co-release are intermingled, with no obvious bias for a distribution along the rostro-caudal or dorsomedial axis by co-release type, or a specific distribution of cells that did not show corelease in the same slices/animals. This result has been added as new Figure 2—figure supplement 3. Of note, we now limited our analysis to animals in which co-release was detected. This resulted in changes in the percentage of cells in which co-release vs. no co-release was detected (Figure 2B), which however does not change our conclusions. Moreover, VTA/SNC injection sites for these animals were similar in location (Figure 2figure supplement 1). In the future and beyond the scope of this study, more precise methods (i.e. intersectional approaches) will be necessary to identify the cell types in the DA midbrain that drive specific response types in amygdala target cell populations.

c. We had reported the fraction of neurons in young and adult mice that were directly hyperpolarized by DA averaged over all bregma-levels. We now retrieved the locations of all neurons that were hyperpolarized by DA or non-responsive for both age groups. We also added an additional new experiment assessing the fraction of neurons hyperpolarized by phasic optogenetic stimulation of DA-afferents in adult animals (see below, response to reviewer 2, new Figure 3). Our consistent observation is that in all three experiments, responsive and non-responsive neurons are intermingled in the dmITC cluster, with no obvious bias for a specific location or bregma level. These results have been added as new Figure 3—figure supplement 2A-C.

2. The authors report that virus injections were localized to the VTA, with less expression in the neighboring SNc. While it is difficult experimentally to isolate these two regions during viral injections, the authors can speculate in the discussion whether the DA input to amygdala subregions from the VTA and SN may play a distinct role in behavior.

We now show exemplary results (replicated in two animals), in which we used dual-color AAV-based tracing from DA-neurons in largely non-overlapping regions of SNC and VTA, and examined their axons in the ITC clusters. We show that both, SNC and VTA send axonal projections to dm-ITCs, but vm-ITCs mostly receive fibers from VTA. We included this new result as Figure 1—figure supplement 5.

The role of the mesolimbic pathway from VTA has been more extensively investigated in different forms of learning and prediction error encoding including fear and extinction behavior. A recent study suggests that medial vs. lateral VTA activity provides a reward prediction error-like signal, and a salience signal, respectively (Cai et al., 2020). Furthermore, emerging evidence is starting to point to a role for the nigrostriatal pathway in emotional regulation. Activation of the SNC for example enhances extinction memory and renders it more resistant to renewal (Bouchet et al., 2018). Therefore, a differential innervation of dm- and vm-ITC clusters from the SNC/lateral VTA may suggest that these inputs onto dm-ITCs have additional roles beyond prediction error signaling during early extinction.

a. Additionally, what is known about the localization of amygdalar-projecting midbrain neurons? Are these predominately in the VTA or SN? Is the expression pattern (e.g., location along A-P or M-L axis) similar for TH/GABA or TH/Glu neurons?

The current knowledge about localization and molecular phenotype of amygdala-projecting midbrain neurons is still incomplete, and due to the lack of specific tools, has remained elusive for ITCs. Amygdala-projecting neurons are localized in both the SNc and VTA, and a distinct population of DA neurons in DR/PAG target CeA (Beier et al., 2015; Bjorklund and Dunnett, 2007; Poulin et al., 2018; Vogt Weisenhorn et al., 2016). A fraction of BLA projecting neurons in the VTA also expresses vGluT2 and/or is localized in medial VTA regions containing TH^+^/vGluT2^+^ neurons (Morales and Margolis, 2017; Poulin et al., 2018). To our knowledge, the only evidence on the source of ITC inputs comes from a 6-OHDA lesion study (Ferrazzo et al., 2019) suggesting that ITCs are targeted by SNc/VTA regions that also target BLA. We have now (see above) provided evidence from dual color anterograde tracing, that dm-ITCs receive afferents from VTA and SNC, while vm-ITCs are mainly targeted by VTA.

We have revised the discussion to address all three points raised here: The source of DA-inputs to ITCs, the molecular phenotype of amygdala-projecting neurons, and putative distinct roles of medial vs. lateral VTA, or SNC inputs in different phases of fear extinction.

3. In what percentage of neurons from Figure 2 tested with glutamate and GABA blockers?

We have included the number of neurons for which we confirmed the categorization by pharmacology in the Figure legend of Figure 2 and Figure 2—figure supplement 2 (n=5 cells from 5 animals for dm-ITCs, n=3 cells from 3 animals for vm-ITCs, n=8 cells from 7 animals for CeA).

Reviewer #2:This manuscript describes the results of experiments exploring the effects of dopaminergic modulation of neurotransmission between GABAergic ITC clusters in the amygdala, attempting to link the observed neuromodulatory mechanisms to fear behavior. In my view, however, there are certain problems with experimental design in this study. My enthusiasm for this work is somewhat diminished for the reasons outlined below.

We regret that this reviewer was less enthusiastic about our work. We have carefully evaluated his/her concerns and now added a significant amount of novel experimental data to address all key criticisms, and to expand and strengthen our results. For clarity, we have re-numbered the issues raised and respond to them point by point.

Most of the functional studies described in the manuscript were performed with exogenously-applied DA. The only results in the paper potentially showing the physiological effects of endogenously-released DA were obtained in experiments assaying an effect of optogenetic stimulation of dopaminergic fibers on sIPSCs in slices. However, the stimulation episode was fairly short (10 pulses at 30 Hz) and I was not able to find any evidence in the manuscript that this stimulation, in fact, resulted in release of DA.

To clarify, the stimulation episode in these experiments was 10 times 10 pulses at 30 Hz

(total of 100 pulses), with which we aimed to mimic physiological bursts of phasic activity in DA midbrain neurons, as stated in the methods. We now provide indirect evidence that DA is released during this stimulation protocol by showing that application of a cocktail of DA-R blockers (D1: SCH23390 10 µM; D2: Sulpiride 20 µM; D4: L-745870 100 nM) fully abolished the decrease in sIPSC amplitude in dm-ITCs. This experiment demonstrates that the effect on sIPSC amplitude was mediated by DA-Rs, and not by other modulatory mechanisms. Furthermore, 1 Hz stimulation with the same number of pulses (100 pulses) did not alter sIPSC amplitude. Together, this suggests, that DA is released upon burst activity from DA midbrain afferents, and endogenous DA release modulates sIPSCs in ITCs. We added these data to a completely restructured Figure 4, and moved the data on sEPSCs to Figure 4-figure supplement 2. We furthermore include confirmation of midbrain injection sites for these experiments in new Figure 4—figure supplement 1.

DA-induced hyperpolarization was observed in a fraction of recorded dm-ITC cells (Figure 5, figure supplement 1B-D) with exogenously-applied DA. The authors could perform analogous measurements, looking at effects of phasic stimulation of DA fibers in the presence of glutamatergic and GABAergic antagonists on membrane polarization in ITCs. If the change in membrane polarization (e.g., hyperpolarization) is observed and it is blocked by the D1 receptor antagonist, it would provide direct evidence that photostimulation did induce DA release in studied projections in quantities sufficient to produce measurable changes in ITC functions.

To address this, we carried out the proposed experiment. Notably, we find that already a single burst of 30 Hz optogenetic stimulation (10 pulses) applied to DA-midbrain afferents in the presence of glutamatergic and GABAergic (GABA_A_ and GABA_B_) antagonists hyperpolarizes dm-ITCs. Importantly, this hyperpolarization was of similar magnitude as that evoked by direct DA application, and was again only observed in a fraction of recorded neurons. These converging results strongly suggests that (1) DA application and optogenetic release of DA are able to induce comparable cellular effects in ITCs, and (2) DA application can serve as a good proxy in other experiments, in which optogenetic stimulation of DA fibers is technically not feasible (see point 4 below). We now attribute a full figure and a figure supplement to show these data (new Figure 3 and Figure 3—figure supplement 1). We moved these data in front of the subsequent experiments that either employ 30 Hz optogenetic stimulation to assess the effect on sIPSCs, or direct DA application to interrogate DA’s role in modulation of optogenetically evoked inhibitory interactions between ITC clusters.

If the amplitude of spontaneous IPCSs is decreased in both dm-ITCs and vm-ITCs following phasic optogenetic activation of dopaminergic fibers, it might be helpful if the authors could speculate how these apparently postsynaptic decreases in GABAergic synaptic currents could be mediated. Moreover, the authors demonstrated nicely that dopaminergic projections to ITCs also release GABA and a fraction of them co-release glutamate. Is there any evidence that the observed effects were mediated by DA but not other neurotransmitters (glutamate or GABA, or both)?

Thank you for pointing this out. As outlined above in point 1, we now provide evidence that the effect on sIPSCs is mediated by DA-Rs. As we observed a full block with DA-R antagonists, we do not believe that other neurotransmitters play a significant role in the modulation of sIPSCs by DA-afferent stimulation (new Figure 4C-D, G).

Regarding the mechanism of modulation, we agree with the reviewer that a change in amplitude would at first glance suggest a postsynaptic mechanism. It may be conceivable, that in ITCs, similarly to what was observed in striatal neurons, DA via DRD1 can also directly modulate GABA_A_-Rs and decrease GABA_A_-currents (Flores-Hernandez et al., 2000; Hoerbelt et al., 2015; Tritsch and Sabatini, 2012). However, we like to point out that we investigated spontaneous IPSCs (sIPSCs), which include action potential-driven events at synapses. Considering that release can be multivesicular (Rudolph et al., 2015), a reduction in presynaptic input activity or presynaptic release probability could contribute to a decrease in sIPSC amplitude. We now address this in the discussion.

The results of experiments showing that DA may modulate inhibitory neurotransmission between ITC clusters (Figure 4) are inconclusive. The reported suppression of IPSCs was observed when DA was applied exogenously in a fairly high concentration (30 uM). As this concentration might be non-physiologically high, it might be necessary to show that DA released endogenously (e.g., in response to photostimulation of dopaminergic fibers with the same stimulation protocol that was used to modulate sIPSCs) can, in fact, suppress GABAergic synaptic transmission between ITC clusters. Again, the authors were able to activate dopaminergic inputs to ITC clusters optogenetically. Thus, it remains unclear why the mentioned experiments with 'endogenous' dopamine were not performed

Firstly, although no data on the concentration of synaptic DA are available for ITCs in the amygdala, there is evidence that DA at synaptic sites in the VTA and striatum can reach peak concentrations in the high µM range, even up to 100 µM at peak (i.e.Courtney and Ford, 2014; Liu et al., 2018). Thus, we do not think that bath application of 30 µM DA yields concentrations in acute slices that are out of the physiological range.

Secondly, we now show that DA application and 30 Hz optogenetic stimulation of DA afferents yields comparable levels of postsynaptic hyperpolarization, suggesting that DA application mimics, and can serve as a good proxy for optogenetically-evoked endogenous DA release.

Thirdly, the proposed experiment is not readily feasible for several reasons:

1. Dual optogenetic stimulation of DA midbrain fibers and ITC clusters is not possible with the available mouse lines, as FoxP2-positive neurons are also present in the midbrain, precluding a specific expression in DA afferents when crossing FoxP2-Cre and DAT-Cre mice.

2. GABAergic co-release from DA midbrain fibers would interfere with analyzing evoked IPSCs between ITC clusters at the same time.

3. Alternative approaches using paired recordings of individually connected dm-ITC→vm-ITC cells would, due the limitations of the slice preparation in preserving individual connectivity, yield a very low success rate, and electrical stimulation would unspecifically activate fibers of passage, precluding reliable conclusions. Both approaches would still have the problem outlined in point 2.

Taking all this into account, we deliberately opted to dissect the effect of DA on modulation of synaptic transmission using exogenous application of DA in combination with specific optogenetic stimulation of inhibitory connections between ITC clusters.

The experiments demonstrating that DA-induced suppression of GABAergic synaptic transmission between ITC clusters may be mediated by D1 receptors are inconclusive for the same reason. The observed inhibitory effect of a specific D1 receptor agonist on evoked IPSCs does not necessarily mean that it could be observed under more physiological conditions. Ideally, these experiments should be performed with endogenously released DA to show that photostimulation-induced suppression of IPSCs between clusters is sensitive to D1 receptor antagonists.

We find this comment to be conceptually flawed, as it seems to be questioning some basic principles of pharmacology. The use of specific receptor agonists is a mainstay in pharmacology, and application of specific receptor agonists to mimic the effect of endogenous ligands is widely accepted to demonstrate the involvement of a particular receptor type in a given physiological process. In this regard, we do not understand why the bath-application of a DRD1 antagonist prior to photostimulation-induced release of DA should be more conclusive than bath application of a DRD1 agonist to explore the receptor subtype underlying a DA-induced suppression of IPSCs between ITC clusters.

However, we have now conducted an experiment using a cocktail of DA-R antagonists rather than one specific antagonist, to block the effect of endogenously released DA on sIPSC suppression (new Figure 4C-D, G).

Taken together, we now provide converging evidence using both antagonists together with endogenous release of DA, and subsequent experiments with specific agonists that strongly support the notion that DA modulates IPSCs, and does so chiefly via DRD1.

The results of extinction experiments are somewhat confusing. The 'early extinction' protocol did not result in any extinction at all. Therefore, the meaning of observed changes in inhibitory synaptic responses in between-clusters connections after extinction training and their sensitivity to dopaminergic modulation is unclear. The observed dissociation between behavioral and network-level effects does not support the notion that the observed neuromodulatory actions of DA on inter-ITC functional connectivity are functionally relevant.

We intended with the short “Early extinction” protocol to use a training protocol that does not yet induce a significant reduction in freezing levels. We aimed to capture a time point for ex vivo recordings when, during initial presentations of the CS without the US, the prediction error that drives learning would be large. It is precisely in this phase, when DA midbrain neurons signal the omission of the US (Cai et al., 2020; Salinas-Hernandez et al., 2018). To achieve effective extinction, comparable studies in mice used longer protocols with more CS presentations in the absence of the US (25-30 CSs on one day) or two sessions over the course of 2 days (Asede et al., 2015; Cai et al., 2020; Senn et al., 2014). This is fully in line with our data presented in Figure 6—figure supplement 2, demonstrating that we can indeed achieve extinction when applying more stimuli on day 2 (reanalysis focusing on within session extinction, revised Figure 6—figure supplement 2), continued training on day 3, and a memory test on day 4.

Reviewer #3:This manuscript investigates how the intercalated cell (ITC) islands of the amygdala are modulated by dopamine, and during extinction. This work is motivated by previous findings demonstrating dopamine release from the VTA during early extinction -when prediction error is high – and dopamine receptor expression on the ITC. The authors first demonstrate that VTA/SNc dopaminergic cells projects to the ITC and co-release GABA, resulting in inhibitory postsynaptic potentials. Using in-vitro recordings, the authors then show that phasic stimulation of dopaminergic input decreases the amplitude of spontaneous inhibitory post-synaptic currents (IPSC) in the dorsomedial and ventromedial ITC. Bath application of dopamine also decreased the amplitudes of evoked IPSCs in the ITC, and increase the Paired Pulse Ratio (PPR) in ITC island-to-island signaling in a D1 receptor dependent manner, suggesting that dopamine mediates presynaptic depression in intra-ITC communication via presynaptic D1 receptor activation. The authors then show that after animals undergo early extinction training, stimulation of dopaminergic inputs from the VTA/SNc weakens the evoked IPSCs in the ITC, and enhances presynaptic depression in ITC-to-ITC signaling relative to tone-only controls. The authors suggest that dopaminergic modulation of intra-ITC signaling, from the dorsomedial to the ventromedial ITC island, mediates disinhibition of the ventromedial ITC, leading to their subsequent inhibition of amygdala output. This work makes an important contribution to the field by providing an interesting mechanism for dopamine-mediated modulation of intra-ITC signaling as a function of extinction learning.

We thank the reviewer for her positive feedback. We were impressed by the thorough assessment of our work carefully scrutinizing our model against the data. We appreciate the constructive input that helped us to improve our manuscript, and hope that the additional data and revisions to the text and model can address all the points raised in a satisfactory fashion.

1. The authors argue that dopaminergic inputs to the ITC have several effects; they weaken spontaneous (and evoked) ITC activity via GABAergic co-release during phasic dopaminergic activity, and weaken dmITC -to-vmITC (and lITC-to-dmITC) signaling by presynaptic depression of this connection. Both of these factors are proposed to contribute to decreased lateral-to-medial inhibition of ITC islands, in particular during periods of early extinction training. Although, this is an elegant model that is supported by the data, the data also suggest that other dopaminergic influences are at play, but are disregarded by the authors' model as inconsequential. For example, the authors find a direct, postsynaptic influence of dopamine on dmITC, whereby 50% of the neurons are directly hyperpolarized by DA. This is described in the text as only a fraction of neurons exhibiting this response (pg. 12, line 256), and the conclusion is made that DA is acting "chiefly by presynaptic DRD1 receptors" (pg. 12, lines 258-9), and again referred to as only a fraction in the Discussion section (pg. 16, line 368). This stress on the presynaptic DRD1 mechanism is also repeated in the model shown in figure 7. Given that half the cells are inhibited by DA post-synaptically, this mechanism should also be incorporated into the model of how DA regulates the ITC.

We agree with the points raised. While we originally intended to simplify our model by focusing on the novel and highly consistent presynaptic mechanism of DA action, we have now also integrated the postsynaptic mechanism into our model (revised Figure 7).

We have extended our work scrutinizing the postsynaptic action of DA by providing evidence that optogenetic 30 Hz stimulation also induces a hyperpolarization, that is of similar size compared to that evoked by direct DA application, and in keeping with previous results, was only observed in a fraction of dm-ITCs (see point 2 of reviewer 2, new Figure 3).

2. Figure 1 shows DA inputs to both dmITC and vmITC. Given that half of dmITC are directly inhibited by DA, it would be useful to know whether the vmITC are also post-synaptically inhibited by DA. The effect of DA on vmITC pertains strongly to the extinction model proposed by the authors, because it predicts that vmITC are released from inhibition via DA through presynaptic action, whereas there it's quite possible that DA directly hyperpolarizes vmITC. It's not clear if these cells would also be inhibited with phasic stimulation (only dmITC results are shown in supplementary figure 5). It's possible that there are fewer inputs from VTA/SNc onto the vmITC than the dmITC, such that even if DA directly inhibits the vmITCs, there aren't as many inputs to them. In the staining shown in figure 1 Supp2, it appears that the VTA/SNc inputs to the dmITC are much stronger than the they are to the vmITC, however this is not quantified, so it's hard to tell if this is just the example. Can the authors quantify DA input fluorescence to the dmITC/ vmITC in order to get an approximation of the strength of DA input to shed light on this question. If DA directly inhibits these cells, then it's likely that they may become more inhibited after extinction training rather than disinhibited, as in the proposed model.

We thank the reviewer for this suggestion and her careful observation. Indeed, a previous study observed DA-induced hyperpolarization in the ventrally-located main ITC nucleus in slices from young mice, which we had referenced in the discussion (Manko et al., 2011). Together with our observation that 30 Hz stimulation of DA-afferents also reduces sIPSCs in vm-ITCs, we certainly agree that DA can also modulate vm-ITCs. We believe, the central point raised that can provide additional support for our model, is a differential DA-midbrain innervation of dm- and vm-ITC clusters. We tackled this using three independent approaches:

1. We assessed the density of dopaminergic VTA/SNC inputs to dm- and vm-ITCs by 3D reconstruction of YFP+ afferents within FoxP2-positive clusters from 4 animals (new Figure 1C-D, new Figure 1—figure supplement 3).

2. To preclude bias due to size and location of injection sites in DAT-Cre mice, we also measured the density of TH^+^ fibers in both dm- and vm-ITC clusters in a separate set of wt animals (n=9, new Figure 1—figure supplement 4A-C).

3. We add exemplary data from dual-color tracing from SNC and VTA obtained from 2 animals (see point 2 of reviewer 1), demonstrating that the dm-ITC cluster is targeted by both, whereas the vm-ITC cluster chiefly receives inputs from VTA (new Figure 1—figure supplement 5).

Together, our converging sets of novel data provide evidence for a differential dopaminergic innervation of these ITC clusters, with a potentially larger and more diverse impact on dm-ITCs. We integrated this aspect in our model (revised Figure 7).

3. Similarly to the point above, another way that this issue could be addressed is to show that vmITC are more excitable post-extinction compared to the controls.

This is an interesting point. However, multiple factors are expected to contribute to ITC activity and excitability (including state-dependent inputs, intrinsic factors, and modulatory influences). As these conditions cannot be recapitulated ex vivo in brain slices. Addressing this question would require single unit in vivo recordings of ITCs during extinction behavior, and would go beyond the scope of this study. However, in support of our model, earlier studies using immediate early gene mapping approaches showed an increased activation of vm-ITCs upon extinction learning and retrieval (Busti et al., 2011).

4. It's not clear what role, if any, the authors believe that the lITC-dmITC connection plays in the model. On the one hand the VTA/SNc connection to the lITC is not assessed neuroanatomically (e.g. it's not shown in figure 1, nor is it drawn in the model in figure 7), or physiologically (e.g. there aren't recordings showing the effects of phasic DA stimulation on lITC islands). On the other hand, the lITC-dmITC connection is shown to undergo PPR depression with DA, similarly to the dmITC-vmITC connection. However, then the question of whether this connection changes with early extinction is not investigated, and not discussed.

We agree with the reviewer that the role of the input from l-ITCs to dm-ITCs is not thoroughly investigated in our study, or well developed in our model. We opted to keep the focus on dm-ITCs, and dm- and vm-ITC connectivity for ex vivo assessment of changes upon behavioral manipulations. We nevertheless believe it is important novel information, that dm-ITCs are under inhibitory control by l-ITCs in the external capsule, which significantly extends the way we think of ITC networks in general. It also drives the point that dm-ITC activity can be modulated by DA in three ways: (1) their inhibitory input and (2) inhibitory output is modulated by presynaptic mechanisms, and (3) a fraction of cells is also inhibited by postsynaptic hyperpolarization. Modulation of these inhibitory inputs onto dm-ITCs could serve to select or sharpen their output, as their downstream targets are diverse (Asede et al., 2015; Busti et al., 2011). We feel however, that a comprehensive investigation of the function and regulation of the l-ITCs themselves would be beyond the scope of this study. Thus, we shifted the l-ITC→dm-ITC dataset to the supplement (new Figure 5—figure supplement 1) to focus on the dm-ITC→vm-ITC projection (revised Figure 5). We also toned down the emphasis on this pathway in our model (revised Figure 7). We mention this triple function of modulation of dm-ITCs by DA, and the implications for ITC network function of the novel connectivity observed from l-ITC→dm-ITC in the discussion.

5. There are a number of instances where the representative examples are confusing because they don't look like the effect that the authors are hoping to demonstrate. For example, in Figure 3 Supp2, the data show that the frequency of sIPSC at baseline and after stimulation doesn't change, whereas the representative example shows that the frequency of the sIPSCs decreases after stimulation. Similarly, in figure 4I, the example traces on the right are meant to show that DA application leads to increased PPR in lITC -to-dmITC synapse, however the PPR traces don't appear to show that phenomenon. The same is true of the lITC-dmITC PPR in figure 5d (DH drug condition doesn't look like the PPR increases, whereas the group data show that it does). The examples and data shown in figures 6b-c are confusing in terms of color (the reference in the legend is a grey color that isn't on the figure), but also in terms of the finding where the extinction and tone controls are mixed up in order.

For clarity, we subdivided the different points raised here in subsections a-c.

Thank you for pointing out these issues.

a. We now show the sIPSCs in new Figure 4 on an enlarged time scale, allowing for better assessment of frequency.

b. All the PPR traces in old Figures 4 and 5 (now Figure 5 and Figure 5—figure supplement 1) were actually representative of the effect size. Please note that in the graphs, we are showing the change in PPR upon DA application and not the absolute PPR values. This includes depressing synaptic connections that undergo an increase in PPR by becoming less depressing, or slightly facilitating.

c. We corrected the color code in Figures 6B-C and 6E-F to correspond to the experimental groups.

6. In some instances, the authors don't report the number of mice from which cells were obtained (e.g. n=6 cells and n=3 cells on pg. 10, lines 211 and 212). Are these cell numbers obtained from multiple mice? It would be helpful to see this data shown as group data, as well as the example data shown.

The number of animals from which the recorded neurons were obtained are now stated throughout.